# PROGFED: EFFECTIVE, COMMUNICATION, AND COMPUTATION EFFICIENT FEDERATED LEARNING BY PROGRESSIVE TRAINING

## ABSTRACT

Federated learning is a powerful distributed learning scheme that allows numerous edge devices to collaboratively train a model without sharing their data. However, training is resource-intensive for edge devices, and limited network bandwidth is often the main bottleneck. Prior work often overcomes the constraints by condensing the models or messages into compact formats, e.g., by gradient compression or distillation. In contrast, we propose ProgFed, the first progressive training framework for efficient and effective federated learning. It inherently reduces computation and two-way communication costs while maintaining the strong performance of the final models. We theoretically prove that ProgFed converges at the same asymptotic rate as standard training on full models. Extensive results on a broad range of architectures, including CNNs (VGG, ResNet, ConvNets) and U-nets, and diverse tasks from simple classification to medical image segmentation show that our highly effective training approach saves up to $20\%$ computation and up to $63\%$ communication costs for converged models. As our approach is also complimentary to prior work on compression, we can achieve a wide range of trade-offs, showing reduced communication of up to $50\times$ at only $0.1\%$ loss in utility.

## 1 INTRODUCTION

Federated Learning (FL) (McMahan et al., 2017) has led to remarkable advances in the development of extremely large machine learning systems. Federated training methods allow multiple clients (edge devices) to jointly train a global model without sharing their private data with others. Despite the progress, training methods in FL still suffer from high communication and computational costs, as edge devices are often equipped with limited hardware resources and limited network bandwidth.

Prior literature has studied various compression techniques to address the computation and communication bottlenecks. We can divide these methods into three main categories (see also Table 1): (i) Compression techniques that represent gradients (or parameters) with fewer bits to reduce communication costs. Prominent examples are quantization (Alistarh et al., 2017; Lin et al., 2018; Fu et al., 2020) or sparsification (Stich et al., 2018; Konečný et al., 2016). (ii) Model pruning techniques that identify (much smaller) sub-networks within the original models to reduce computational cost at inference (Li & Wang, 2019; Lin et al., 2020). And (iii) knowledge distillation (Hinton et al., 2015) techniques, that allow the server to distill the knowledge from the clients with hold-out datasets (Li & Wang, 2019; Lin et al., 2020). However, these methods can sometimes degrade performance.

Progressive learning (Karras et al., 2018) is a well-known technique that has been used in image generation to stabilize the training. The primary idea is to first train the shallower layers on simpler tasks (e.g., images with lower resolution) and gradually grow the network to tackle more complicated tasks (e.g., images with higher resolution). By applying progressive training in a federated learning setting, we can benefit from highly reduced resource demands (the computation and communication load is drastically reduced when training shallow models). By growing the models to their original size, we can recover (and often surpass) the accuracies reached by standard training. Moreover, we observe that progressive training facilitates the learning process and stabilizes training. Despite the appealing features, no previous study has systematically investigated exploiting progressive learning to reduce the costs in federated learning.

Table 1: Comparison of ProgFed to other compression schemes.

| Technique | Computation Reduction | Communication Reduction | Dataset Efficiency |
|---|---|---|---|
| Message Compression | | ✓ | ✓ |
| Model Pruning | ✓ (only for inference) | | ✓ |
| Model Distillation | ✓ | ✓ | |
| ProgFed (Ours) | ✓ | ✓ | ✓ |

In this work, we propose ProgFed, the first progressive learning framework that reduces both communication and computation costs in FL. In our approach, we divide the model into several overlapping partitions and introduce lightweight local monitoring heads to guide the training of the submodels. During training, the model capacity is gradually increased, until it reaches the full model of interest. This, of course, reduces the computation and communication costs, since the shallow sub models have much fewer parameters than the full model. We show theoretically that ProgFed training converges at the same asymptotic rate as standard training on full models. Due to the nature of progressive learning, our method can reduce computational overheads and provides two-way communication savings (both from server-to-client and client-to-server directions). We demonstrate that our method can reach impressive performance (matching the baselines) using much fewer communication. ProgFed is compatible with classical compression—including quantization and sparsification—which can be applied on top of progressive training. We experimentally show that combing these features enables a higher compression ratio and may motivate more advanced compression schemes based on progressive learning.

We summarize our main contributions as follows.

- We propose ProgFed, the first progressive learning framework to reduce the training resource demands (computation and two-way communication). We theoretically prove that ProgFed converges at the same asymptotic rate as standard training the full model.
- We conduct extensive experiments on various datasets (CIFAR-10/100, EMNIST and BraTS) and architectures (VGG-16/19, ResNet-18/152, ConvNets, 3D-Unet) to show that with the same number of epochs, our method can save around $25\%$ computation cost, up to $32\%$ two-way communication costs in federated classification, and $63\%$ in federated segmentation without sacrificing performance.
- In addition, our method helps the models save around $2\times$ fewer communication costs in classification and $6.5\times$ fewer costs in U-net segmentation to achieve practicable performance ($\geq 98\%$ of the best baseline). This is beneficial for combating limited training budgets in federated learning.
- Lastly, we show that our method complements classical compression and appears robust against compression errors. It permits a higher compression ratio and may motivate more advanced compression schemes based on progressive learning. With these combined techniques, we are able to reduce communication of up to $50\times$ at only $0.1\%$ loss in utility.

## 2 RELATED WORK

In this section, we review work on cost reduction and leave progressive learning in Section D.

**Message compression.** Fruitful literature has studied message compression (e.g. on gradients or model weights) to reduce the communication costs in distributed learning. The first category focuses solely on client-to-server compression (Alistarh et al., 2017; Wen et al., 2017; Lin et al., 2018; Bernstein et al., 2018; Stich et al., 2018; Konečnỳ et al., 2016; Karimireddy et al., 2019; Fu et al., 2020; Stich, 2020). To name a few, Konečnỳ et al. (2016) reduce the costs by sending sparsified gradients and compressing them with probabilistic quantization. Alistarh et al. (2017); Wen et al. (2017) prove the convergence of probabilistic quantized SGD. SignSGD (Bernstein et al., 2018) significantly compresses the gradients with only one bit, while Karimireddy et al. (2019) fix the biased nature of SignSGD with an error-feedback mechanism and generalize to other compression schemes. Compression on server-to-client communication has been shown non-trivial and attracted many recent focuses (Yu et al., 2019; Tang et al., 2019; Liu et al., 2020; Philippenko & Dieuleveut, 2020). Instead of dedicated designs, our method inherently reduces two-way communication costs and complements existing methods as we will show in Section 4.

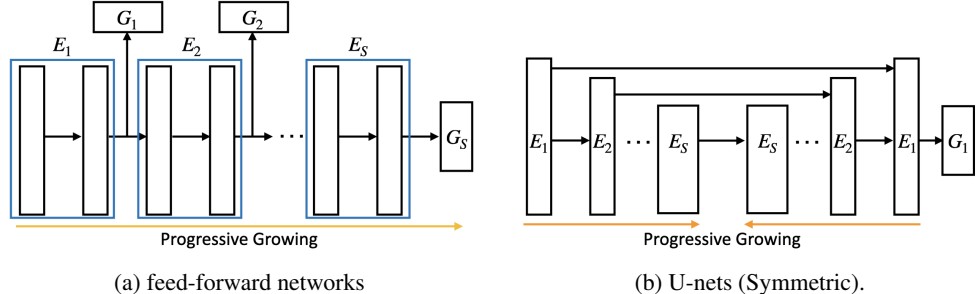

(a) feed-forward networks       (b) U-nets (Symmetric).

Figure 1: An overview of ProgFed on (a) feed-forward networks and (b) U-nets (Symmetric). We progressively train a deep neural network from the shallower sub-models, e.g. $\mathcal{M}^1$ consisting of main block $E_1$ and head $G_1$ (Eq. 2), gradually expanding to the full model $\mathcal{M}^S = \mathcal{M}$ (Eq. 1). Note that the local heads $G_i$ in feed-forward networks are only used for training sub-models and discarded when progressing to the next stage.

**Model pruning.** Model pruning removes redundant weights to address the resource constraints (Mozer & Smolensky, 1989; LeCun et al., 1990; Frankle & Carbin, 2019; Lin et al., 2019). There are two categories: *unstructured* and *structured* pruning. *Unstructured* methods prune individual model weights according to certain criteria such as Hessian of the loss function (LeCun et al., 1990; Hassibi & Stork, 1993) and small magnitudes (Han et al., 2015). However, these methods cannot fully accelerate without dedicated hardware since they often result in sparse weights. In contrast, *structured* pruning methods prune channels or even layers to alleviate the issue. They often learn importance weights for different components and only keep relatively important ones (Liu et al., 2017; Yu et al., 2018; Mohtashami et al., 2021; Li et al., 2021). Despite the efficiency, model pruning usually happens at inference and does not reduce training costs while our method achieves computation- and communication-efficiency even during training.

**Model distillation.** Another line of research for model compression is model distillation (Buciluǎ et al., 2006; Hinton et al., 2015), which requires a student model to mimic the behavior of a teacher model (Polino et al., 2018; Sun et al., 2019). Recent work has investigated transmitting logits rather than gradients (Li & Wang, 2019; Lin et al., 2020; He & Annavaram, 2020; Choquette-Choo et al., 2020), which significantly reduces communication costs. However, these methods either require additional query datasets (Li & Wang, 2019; Lin et al., 2020) or cannot enjoy the merit of datasets from different sources (Choquette-Choo et al., 2020). In contrast, our work reduces the costs while retaining the dataset efficiency.

## 3 PROGFED

Federated learning is a distributed learning framework in which a huge amount of clients train local models independently, and a central server aggregates the client updates to learn a global model. Motivated by progressive learning, we propose ProgFed that progressively expands the network from a shallower one to the complete model. Our method is effective and efficient for multi-client training while we consider only one client in theoretical analysis for conciseness. The proposed model splitting and progressive growing are illustrated in Figure 1, and the federated optimization scheme is summarized in Algorithm 1. We now present the proposed method in detail below.

### 3.1 PROPOSED METHOD

**Model Structure.** We now describe the proposed training method. For a given a machine learning model $\mathcal{M}$, i.e. a function $\mathcal{M}(\cdot, \mathbf{x}): \mathbb{R}^n \to \mathbb{R}^k$ that maps $n$-dimensional input to $k$ logits for parameters (weights) $\mathbf{x} \in \mathbb{R}^d$, we assume that the network can be written as a composition of blocks (feature extractors) $E_i$ along with a task head $G_S$, namely,

$$\mathcal{M} := G_S \circ \bigcirc_{i=1}^{S} E_i = G_S \circ E_S \circ \cdots \circ E_2 \circ E_1. \tag{1}$$

Note that the $E_i$'s could denote e.g., a stack of residual blocks or simply a single layer. The learning task is solved by minimizing a loss function of interest $\mathcal{L} : \mathbb{R}^k \to \mathbb{R}$ (e.g., cross-entropy) that maps the predicted logits of $\mathcal{M}$ to a real value, i.e. minimization of $f(\mathbf{x}) := \mathcal{L} \circ \mathcal{M}(\mathbf{x})$.

**Progressive Model Splitting.** To achieve progressive learning, we first divide the network $\mathcal{M}$ into $S$ stages, denoted by $\mathcal{M}^s$, for $s \in \{1, \dots, S\}$ associated with the split indices. We additionally introduce local supervision heads for providing supervision signals. Formally, we define

$$\mathcal{M}^s := G_s \circ \bigcirc_{i=1}^{s} E_i \,, \tag{2}$$

where $G_s$ is a newly introduced head. Each head $G_s$, for $s < S$ consists of only a pooling layer and a fully-connected layer in our experiments for feed-forward networks. The motivation is that simpler heads may encourage the feature extractors $E_i$ to learn more meaningful representations. Note that the sub-model $\mathcal{M}^s : \mathbb{R}^n \to \mathbb{R}^k$ produces the same output size as the desired model $\mathcal{M}$; therefore, its corresponding loss $f^s(\mathbf{x}^s) := \mathcal{L} \circ \mathcal{M}^s(\mathbf{x}^s)$ can be trained with the same loss criterion $\mathcal{L}$ as the full model.

**Training of Progressive Models.** We propose to train each sub-model $\mathcal{M}^s$ for $T_s$ iterations (a certain fraction of the total training budget) and gradually grow the network from $\mathcal{M}^1$ to $\mathcal{M}^S = \mathcal{M}$. When growing the network from stage $s$ to $s + 1$, we pass the corresponding parameters of the pretrained blocks $E_i$, $i \leq s$, to the next model $\mathcal{M}^{s+1}$ and initialize its blocks $E_{s+1}$ and $G_{s+1}$ with random weights. Once the progressive training phase is completed, we continue training the full model $\mathcal{M}$ in an end-to-end manner for the remaining iterations. The length $T_s$ of each progressive training phase is a parameter that could be fine-tuned individually for each stage (depending on the application) for best performance. However, as a practical guideline that we adopted for all experiments in this paper, we found that denoting roughly half of the total number of training iterations $T$ to progressive training, and setting $T_s = \frac{T}{2S}$ for $s < S$, $T_S = \frac{T(S+1)}{2S}$, such that $T = \sum_{s=1}^{S} T_s$, works well across all considered training tasks.

**Extension to U-nets.** In addition to feed-forward networks (Figure 1(a)), we show that our method can generalize to U-net (Figure 1(b)). U-net typically consists of an encoder and a decoder. Unlike feed-forward networks, the encoder sends both the final and intermediate features to the decoder. Therefore, we propose to grow the network from outer to inner layers as shown in Figure 1(b) and retain the original output heads as $G_i$. We refer to the strategy as the *Symmetric* strategy. In contrast, we propose another baseline, the *Asymmetric* strategy, which adopts the full encoder at the beginning and gradually grows until it reaches the full decoder. For this strategy, we also adopt several temporal heads for earlier training stages. As we will show in Section 4.3, the *Symmetric* strategy significantly outperforms the *Asymmetric* strategy, which supports the notion of progressive learning.

**Practical considerations.** We empirically observe that learning rate restart (Loshchilov & Hutter, 2017) clearly facilitates training in the centralized setting. This is because sub-models may overfit the local supervision while learning rate restart helps the sub-models with the newly added layers escape from the local minima and quickly converge to a better minima. On the other hand, warm-up (Goyal et al., 2017) for the new layers plays an important role in federated learning. Model weights often take a longer time to converge in federated learning, which makes the newly added layers introduce more noise to the previous layers. With warm-up, the new layers recover the performance without affecting previous layers. For instance, the existence of warm-up leads to around 2% difference (53.23 vs. 51.09) on CIFAR-100 with ResNet-18.

## 3.2 Theoretical Analysis

In this section we prove that progressing training converges to a stationary point at the rate $\mathcal{O}\left(\frac{1}{\epsilon^2}\right)$, i.e. with the same asymptotic rate as SGD training on the full network. For this, we extend the analysis from (Mohtashami et al., 2021) that analyzed the training of partial subnetworks. However, in our case the networks are not subnetworks, $\mathcal{M}^s \not\subset \mathcal{M}^{s+1}$ (as the head is not shared), and we need to extend their analysis to progressive training with different heads.

**Notation.** We denote by $\mathbf{x}^s$ the parameters of $f^s$, $s \in \{1, \dots, S\}$ and abbreviate $\mathbf{x}^S = \mathbf{x} \in \mathbb{R}^d$ for convenience. For $s \leq i \leq S$ let $x^i_{|E_s}$ and $\nabla f^i(\mathbf{x}^i)_{|E_s}$ denote the projection of the parameter $\mathbf{x}^i$ and gradient $\nabla f^i(\mathbf{x}^i)$ to the dimensions corresponding to the parameters of $E_1$ to $E_s$ (without parameter of $E_{s+1}$ to $E_i$ and without head $G_i$). In iteration $t$, the progressive training procedure updates the parameters of model $f^s$, $s = \min(S, \lceil \frac{t}{T_s} \rceil)$. For convenience, we do not explicitly write the dependency of $s$ on $t$ below, and use the shorthand $x^s_t$ to denote the corresponding model at timestep $t$. We further define $\mathbf{x}_t$ such that $\mathbf{x}_{t|E_s} = x^t_{s|E_s}$ and $\mathbf{x}_{t|E_s^\complement} = \mathbf{x}_{0|E_s^\complement}$ on the complement.

---

**Algorithm 1** ProgFed—Progressive training in a Federated Learing setting

---

1: **Input:** initialization $\mathbf{x}_0^1$, model $\mathcal{M}(\cdot, \mathbf{x}_0)$, iteration budgets $T$, $T_s$, number of stages $S$, $s = 1$, desired number of local updates $J \geq 1$, learning rate $\eta$
2: **Output:** parameters $\mathbf{x}_T$ and trained model $\mathcal{M}(\cdot, \mathbf{x}_T)$
3: **for** $t = 1, \ldots, T$ **do**
4:     **if** $\min(S, \lceil \frac{t}{T_s} \rceil) > s$ **then**                ▷ switch from $\mathcal{M}^s$ to $\mathcal{M}^{s+1}$ after $T_s$ iterations
5:         initialize parameter $\mathbf{x}_t^{s+1}$ randomly      ▷ initialize new block $E_{s+1}$ and new head $G_{s+1}$
6:         $\mathbf{x}_{t|E_s}^{s+1} \leftarrow \mathbf{x}_{t|E_s}^s$                  ▷ copy parameters of shared blocks $E_1, \ldots, E_s$.
7:         $s \leftarrow s + 1$                           (the old head $G_s$ is discarded)
8:     **end if**
9:     Sample a subset $\mathcal{C}$ of clients         ▽ standard Federated Learning on active model $\mathcal{M}^s$
10:     **for** each active client $c \in \mathcal{C}$ **do**
11:         initialize $\mathbf{x}_{c,1}^s \leftarrow \mathbf{x}_t^s$                  ▷ send $\mathbf{x}_t^s$ to active clients
12:         **if** warm-up is needed after growing **then**
13:             freeze $\mathbf{x}_{c,1|E_{s-1}}^s$ and warm-up $\mathbf{x}_{c,1|E_{s-1}^\complement}^s$     ▷ warm up the newly added layers
14:         **end if**
15:         **for** $j = 1, \ldots, J$ **do**                      ▽ Local SGD updates
16:             $\mathbf{x}_{c,j+1}^s = \mathbf{x}_{c,j}^s - \eta g_c^s(\mathbf{x}_{c,j}^s)$    ▷ compute (mini-batch) gradient $g_c^s$ on client $c$'s data
17:         **end for**
18:         $\Delta_c = \mathbf{x}_{c,J} - \mathbf{x}_{c,1}$
19:     **end for**
20:     $\mathbf{x}_{t+1}^s = \mathbf{x}_t^s + \frac{1}{|\mathcal{C}|} \sum_{c=1}^{\mathcal{C}} \Delta_c$             ▷ aggregate updates from the clients
21: **end for**

---

**Assumption 1** ($L$-smoothness). The function $f \colon \mathbb{R}^d \to \mathbb{R}$ is differentiable and there exists a constant $L > 0$ such that

$$\|\nabla f(\mathbf{x}) - \nabla f(\mathbf{y})\| \leq L \|\mathbf{x} - \mathbf{y}\|. \tag{3}$$

We assume that for every input $\mathbf{x}^s$, we can query an unbiased stochastic gradient $g^s(\mathbf{x}^s)$ with $\mathbb{E}[g^s(\mathbf{x}^s)] = \nabla f^s(\mathbf{x}^s)$. We assume that the stochastic noise is bounded.

**Assumption 2** (bounded noise). There exist a parameter $\sigma^2 \geq 0$ such that for any $s \in \{1, \ldots, S\}$:

$$\mathbb{E} \|g^s(\mathbf{x}^s) - \nabla f^s(\mathbf{x}^s)\|^2 \leq \sigma^2, \qquad \forall \mathbf{x^s}. \tag{4}$$

The progressive training updates $\mathbf{x}_{t+1}^s = \mathbf{x}_s^t - \gamma_t g^s(\mathbf{x}_t^s)$ with a SGD update on the model $\mathbf{x}_t^s$. With the two assumptions above, which are standard in the literature, we prove the convergence of submodels $\mathcal{M}^s$ as well as the model of interest $\mathcal{M}$.

**Theorem 1.** Let Assumptions 1 and 2 hold, and let the stepsize in iteration $t$ be $\gamma_t = \alpha_t \gamma$ with $\gamma = \min \left\{ \frac{1}{L}, (\frac{F_0}{\sigma^2 T})^{\frac{1}{2}} \right\}$, $\alpha_t = \min \left\{ 1, \frac{\langle \nabla f(\mathbf{x}_t)_{|E_s}, \nabla f^s(\mathbf{x}_t^s)_{|E_s} \rangle}{\|\nabla f^s(\mathbf{x}_t^s)_{|E_s}\|^2} \right\}$. Then it holds for any $\epsilon > 0$,

- $\frac{1}{T} \sum_{t=0}^{T-1} \alpha_t^2 \|\nabla f^s(\mathbf{x}_t^s)_{|E_s}\|^2 < \epsilon$, after at most the following number of iterations T:

$$\mathcal{O}\left( \frac{\sigma^2}{\epsilon^2} + \frac{1}{\epsilon} \right) \cdot L F_0. \tag{5}$$

- Let $q := \max_{t \in [T]} \left( q_t := \frac{\|\nabla f(\mathbf{x}_t)\|}{\alpha_t \|\nabla f^s(\mathbf{x}_t^s)_{|E_s}\|} \right)$, then $\frac{1}{T} \sum_{t=0}^{T-1} \|\nabla f(\mathbf{x}_t)\|^2 < \epsilon$ after at most the following iterations $T$:

$$\mathcal{O}\left( \frac{q^4 \sigma^2}{\epsilon^2} + \frac{q^2}{\epsilon} \right) \cdot L F_0, \tag{6}$$

where $F_0 := f(\mathbf{x}_0) - (\min_{\mathbf{x}} f(\mathbf{x}))$.

Theorem 1 shows the convergence of the full model $\mathcal{M}$. The convergence is controlled by two factors, the alignment factor $\alpha_t$ and the norm discrepancy $q_t$. The former term measures the similarity between the corresponding parts of the gradients computed from the sub-models and the full model (note that $\alpha_t \equiv 1$ in the last phase of training on $f^S = f$). The latter term $q$ measures the magnitude discrepancy (in Figure 7 we display the evolution of $q_t$ during training for one example task, note

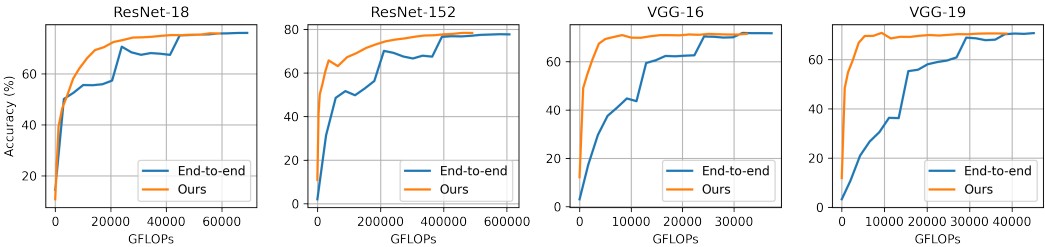

Figure 2: Accuracy (%) vs. GFLOPs on CIFAR-100 in the centralized setting.

that $q_t = 1$ in the last phase of training). We would like to highlight that the convergence criterion in the first statement is lower bounded by the average gradient in the last phase of the training, $\frac{1}{2} \cdot \frac{1}{2T} \sum_{t=T/2}^{T-1} \|\nabla f(\mathbf{x}_t)\|^2 \leq \frac{1}{T} \sum_{t=0}^{T-1} \alpha_t^2 \|\nabla f^s(\mathbf{x}_t^s)_{|E_s}\|^2$ (this is due to our choice of the length of the phases, with $T_S \geq T/2$). This means, that progressive training will provably require at most twice as many iterations but can reach the performance of SGD training on the full model with much cheaper per-iteration costs.

## 4 EXPERIMENTS

### 4.1 SETUP

We now describe the main implementation details and provide supplementary details in Section B.

**Datasets, tasks, and models.** We consider four datasets: CIFAR-10 (Krizhevsky et al., 2009), CIFAR-100 (Krizhevsky et al., 2009), EMNIST (Cohen et al., 2017), and BraTS (Menze et al., 2014; Bakas et al., 2017; 2018). The former three are for image classification, while the last one is a medical image dataset for tumor segmentation. We conduct centralized experiments for analyzing the basic properties of our method while considering practical applications in federated settings. For the centralized settings, we train VGG-16, VGG-19 (Simonyan & Zisserman, 2014), ResNet-18, and ResNet-152 (He et al., 2016) on CIFAR-100 (100 clients, IID). For the federated settings, we train ConvNets on CIFAR-10 and EMNIST (3400 clients, non-IID), ResNet-18 on CIFAR-100 (500 clients, non-IID), and 3D-Unet (Sheller et al., 2020) on the BraTS dataset (10 clients, IID). Note that we follow Hsieh et al. (2020) to replace batch normalization in ResNet-18 with group normalization.

**Implementation.** We implement all settings with Pytorch (Paszke et al., 2019). For the experiments in the centralized settings, we implement models based on DeVries & Taylor (2017), where we run all experiments for 200 epochs and decay the learning rates in {*60, 120, 160*} epochs by a factor of 0.1. We additionally adopt warm-up (Goyal et al., 2017) and learning rate restart (Loshchilov & Hutter, 2017) in our method to better fit in progressive learning. For federated classification, we follow federated learning benchmarks in (McMahan et al., 2017; Reddi et al., 2021) to implement CIFAR-10, CIFAR-100, and EMNIST, respectively. For federated tumor segmentation, we follow Sheller et al. (2020) for the settings and data splits. We set $S = 3$ for EMNIST and $S = 4$ for all the other datasets and $T_s$ as the practical guideline described in Section 3. We adopt 5 and 25 warm-up epochs for federated EMNIST and federated CIFAR-100, respectively. We summarize the parameters (and number of clients) in Table 6.

### 4.2 COMPUTATION EFFICIENCY

We first analyze the computation efficiency of our method in the centralized setting (where all data is available on a single device) to study the effect of the progressive training in isolation, before moving to the federated use cases. We average the outcomes over three random seeds and we consider four architectures on CIFAR-100, including VGG-16, VGG-19, ResNet-18, and ResNet-152. As shown in Table 2, our method performs comparably to the baselines (that train on the full model) after 200 epochs while consuming fewer floating-point operations per second (FLOPs) and training wall-clock time.

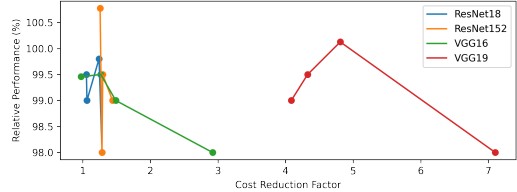 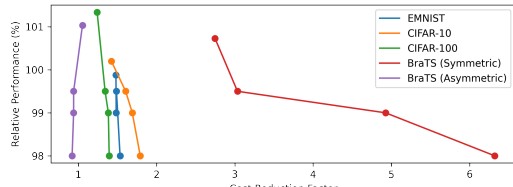

Figure 3: Computation cost reduction at $\{98\%, 99\%, 99.95\%, best\}$ compared to the baseline (training full models) performance in the centralized setting.

Figure 4: Communication cost reduction at $\{98\%, 99\%, 99.95\%, best\}$ compared to the baseline performance in the federated setting.

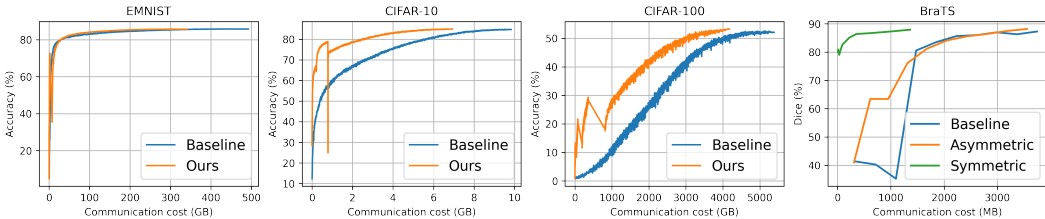

Figure 5: Communication cost vs. Accuracy (%) in federated settings on EMNIST (3400 clients, non-IID), CIFAR-10 (100 clients, IID), CIFAR-100 (500 clients, non-IID), and BraTS (10 clients, IID).

To analyze the efficiency, we report the performance when consuming different levels of costs. Figure 2 shows that our method (orange lines) consistently lies above end-to-end training on the full model (blue lines), meaning that our method consumes fewer computation resources to improve the models. Moreover, we visualize $\{98\%, 99\%, 99.95\%, best\}$ of the performance of the converged baseline (analysis with a larger range is presented in Figure 12). Figure 3 indicates that our method improves computation efficiency across architectures. In the best case, our method can accelerate training up to $7\times$ faster when considering limited computation budgets. We also observe that VGG models improve more than ResNets. A possible reason might be that due to local supervision, sub-models enjoy larger gradients compared to end-to-end training, while it rarely benefits ResNets since skip-connections could partially avoid the problem.

## 4.3 COMMUNICATION EFFICIENCY

Table 2: Results on CIFAR-100 in the centralized setting.

|  | Accuracy | | Reduction | |
| --- | --- | --- | --- | --- |
|  | End-to-end | Ours | Walltime | FLOPs |
| ResNet-18 | **76.08 ± 0.12** | 75.84 ± 0.28 | -24.75% | -14.60% |
| ResNet-152 | 77.77 ± 0.38 | **78.57 ± 0.33** | -22.75% | -19.68% |
| VGG16 (bn) | **71.79 ± 0.15** | 71.54 ± 0.45 | -14.57% | -13.02% |
| VGG19 (bn) | 70.81 ± 1.18 | **70.90 ± 0.43** | -22.10% | -14.43% |

We experiment in the federated setting to verify the communication efficiency of our method. In particular, we consider classification tasks on three datasets, EMNIST, CIFAR-10, and CIFAR-100, and tumor segmentation tasks on the BraTS dataset. We follow the standard protocol as described in Section 4.1 to train the models and average the results over three random seeds. Results in Table 3 indicate that our method achieves comparable results on EMNIST and outperforms the baselines on all the other datasets. In addition, our method saves 20% to 30% two-way communication costs in classification and up to 63% costs in segmentation. The result simultaneously confirms the effectiveness and efficiency of our method.

We compare the performance at different communication costs in Figure 5. We observe that our method is communication-efficient over every cost budget, especially when the model parameters are not evenly distributed across sub-models. For instance, 3D-Unet has most of its parameters in the middle part of the model, making our *Symmetric* update strategy extremely efficient. On the other hand, the *Asymmetric* strategy shows marginal improvement since it starts from the heaviest portion of the model. The finding aligns with the motivation of progressive learning: learning from simpler models might facilitate training. More analysis on segmentation is in Section C.4.

Table 4: Federated ResNet-18 on CIFAR-100 with compression. LQ-X denotes linear quantization followed by used bits representing gradients, and SP-X denotes sparsification followed by the percentage of kept gradients (See Table 10 for standard deviations).

| | Float | LQ-8 | LQ-4 | LQ-2 | SP-25 | SP-10 | LQ-8 +SP-25 | LQ-8 +SP-10 |
|---|---|---|---|---|---|---|---|---|
| | | | | Accuracy | | | | |
| Baseline | 52.54 | 49.40 | 49.55 | 47.26 | 51.23 | 51.79 | 49.67 | 50.25 |
| Ours | **53.23** | **53.07** | **52.32** | **52.87** | **52.00** | **51.86** | **52.19** | **52.24** |
| | | | | Compression Ratio (%) | | | | |
| Baseline | 100 | 25.00 | 12.50 | 6.25 | 25.00 | 10.00 | 6.25 | 2.50 |
| Ours | **77.10** | **19.28** | **9.64** | **4.82** | **19.28** | **7.71** | **4.82** | **1.93** |

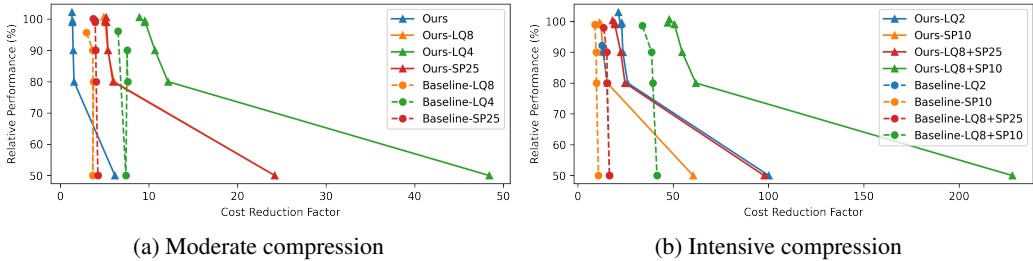

(a) Moderate compression  (b) Intensive compression

Figure 6: Relative performance vs. communication cost reduction with federated ResNet-18 on CIFAR-100 with (a) modest compression and (b) intensive compression.

Table 3: Results in federated settings. We report accuracy (%) for classification and Dice scores (%) for segmentation, followed by cost reduction (CR) as compared to the baselines.

| | Baseline | Ours | CR |
|---|---|---|---|
| EMNIST | **85.75 ± 0.11** | 85.67 ± 0.06 | -29.49% |
| CIFAR-10 | 84.67 ± 0.14 | **84.85 ± 0.30** | -29.70% |
| CIFAR-100 | 52.54 ± 0.44 | **53.23 ± 0.09** | -22.90% |
| BraTS (Aym.) | 86.77 ± 0.45 | **87.66 ± 0.49** | -5.02% |
| BraTS (Sym.) | 86.77 ± 0.45 | **87.96 ± 0.03** | -63.60 % |

Lastly, we analyze the cost reduction when achieving $\{98\%, 99\%, 99.5\%, best\}$ of the performance of the converged baseline. This experiment studies the behavior of our method when only granted limited budgets. Figure 4 (and Figure 13 in the appendix that displays a larger range) presents that except for the *Asymmetric* strategy, our method improves communication efficiency across all datasets. In particular, it achieves practicable performance with 2x fewer costs in classification and up to 6.5x fewer costs in tumor segmentation. We also observe that the communication efficiency improves more when considering lower budgets. This property benefits when the time and communication budgets are limited (McMahan et al., 2017).

### 4.4 COMPATIBILITY

We show that our method complements classical compression techniques including quantization and sparsification. We train several ResNet-18 on CIFAR-100 in the federated setting and apply linear quantization and sparsification following Konečný et al. (2016). Specifically, we consider 8 bits, 4 bits, and 2 bits for quantization (denoted by LQ-X), 25% and 10% for sparsification (denoted by SP-X), and their combinations. Table 4 demonstrates the results of our method and the baseline equipped with various compression techniques. Our method clearly outperforms the baselines in all settings. It indicates that our method is more robust against compression errors, compatible with classical techniques, and thus permits a higher compression ratio.

In addition, we visualize $\{50\%, 80\%, 90\%, 99\%, 99.5\%, best\}$ of relative performance against communication cost reduction in Figure 6. We observe that our method is more efficient across all percentages in every pair (Ours vs. Baseline, plotted in the same color). Besides, the baseline fails to achieve comparable performance in many settings, e.g., the ones with quantization, while our

Table 5: Comparison between update strategies on CIFAR-100 with ResNet-18.

|  | Baseline | Ours | Layerwise | Partial | Mixed | Random |
|---|---|---|---|---|---|---|
| Acc (%) | **76.08±0.12** | 75.84±0.28 | 72.40±0.16 | 74.70±0.04 | 75.04±1.26 | 74.38±0.97 |
| Ratio | 1 | **0.86** | 1 | 1 | ≈ 1 | 0.88 |

method retains comparable performance even with high compression ratios. Interestingly, even with additional compression, our method still facilitates learning at earlier stages. For example, Ours-LQ8+SP25 achieves comparable performance around 50x faster than the baseline, 60x faster to achieve 80%, and more than 200x faster to achieve 50% of performance. Overall, these properties grant our method to adequately approach limited network bandwidth and open up the possibility of more advanced compression schemes.

## 4.5 ANALYSIS OF PROGFED

**Effect of norm discrepancy.** As discussed in Section 3.2, the convergence rate of the full model is controlled by norm discrepancy, namely $q$. As $q_t$ approaches 1, the convergence rate will be closer to the convergence speed of the sub-models. We empirically evaluate the norm discrepancy on CIFAR-100 with ResNet-18 in the centralized setting. Figure 7 shows that the norm discrepancy decreases as the sub-models gradually recover the full model. It suggests that spending too much time on earlier stages may hurt the convergence speed while it offers a higher compression ratio. This outlines the trade-off between communication and training efficiency.

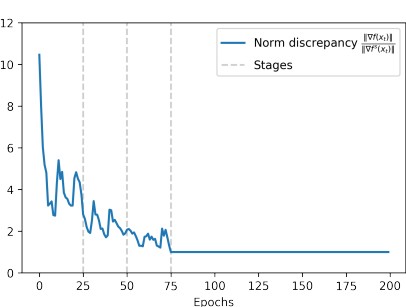

Figure 7: Norm discrepancy.

**Comparison between update strategies.** As described in Section 3, ProgFed progressively trains the network from the shallowest sub-model $\mathcal{M}^1$ to the full model $\mathcal{M}$. We verify our update strategy by comparing it with various baselines in the centralized setting. `Baseline`: end-to-end training; `Layerwise` only updates the latest layer $E_i$ while still passing the input through the whole model $\mathcal{M}$; `Partial` is similar to our method but acquires supervision signals from the last head $G_S$; `Mixed` combines `Partial` and `Ours`, trained on supervision from both $G_i$ and $G_S$; `Random` randomly chooses a sub-model $\mathcal{M}^s$ to update, rather than follows progressive learning.

Table 5 presents the performance and the computation cost ratio. We make several crucial observations. (1) `Ours` and `Random` do not pass the input through the whole network, making them consume fewer computation costs. (2) `Layerwise` greedily trains the network but achieves the worst performance, which highlights the importance of end-to-end fine-tuning. (3) `Ours` outperforms `Random` in both costs and accuracy, verifying the necessity of progressive learning. We also note that our method does not require additional memory space (compared to `Random`) and is easy to implement.

## 5 CONCLUSION

Beyond prior work on expressing models in compact formats, we show a novel approach to modifying the training pipeline to reduce the training costs. We propose ProgFed and show that progressive learning can be seamlessly applied to federated learning for communication and computation cost reduction. Extensive results on different architectures from small CNNs to U-Net and different tasks from simple classification to medical image segmentation show that our method is communication- and computation-efficient, especially when the training budgets are limited. Interestingly, we found that a progressive learning scheme has even led to improved performance in vanilla learning and more robust results when learning is perturbed e.g. in the case of gradient compression, which highlights progressive learning as a promising technique in itself with future application domains such as privacy-preserving learning and advanced compression schemes.

REPRODUCIBILITY STATEMENT

We have provided all assumptions in the main paper and attached the proof of Theorem 1 in the appendix. All implementation details and datasets have been discussed in the main paper and the appendix. We attached the source code as a supplementary material and will publicly release the source code upon acceptance.

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

# A APPENDIX

## A.1 PROOF OF THEOREM 1

In this section we prove Theorem 1. The proof builds on (Mohtashami et al., 2021) that considered training of subnetworks, but not the progressive learning case.

**Lemma 2.** Let $\mathbf{x}_t$ denote the weights of the full model, and $\mathbf{x}_t^s$ the weights of the model that is active in iteration $t$. Note that it holds $\mathbf{x}_{t|E_s} = \mathbf{x}_{t|E_s}^s$ as per the definition in the main text. It holds,

$$\mathbb{E}f(\mathbf{x}_{t+1}) \leq f(\mathbf{x}_t) - \frac{\gamma}{2}\alpha_t^2 \left\|\nabla f^s(\mathbf{x}_t^s)_{|E_s}\right\|^2 + \frac{\gamma^2 L}{2}\sigma^2 \tag{7}$$

*Proof.* Let's abbreviate $\mathbf{g}_t^s = g^s(\mathbf{x}_t^s)$. By the update equation $\mathbf{x}_{t+1}^s = \mathbf{x}_t^s - \gamma_t \mathbf{g}_t^s$ it holds $\mathbf{x}_{t+1|E_s} = \mathbf{x}_{t|E_s} - \gamma_t \mathbf{g}_{t|E_s}^s$. With the $L$-smoothness assumption and the definition of $\alpha_t$,

$$\mathbb{E}f(\mathbf{x}_{t+1}) \leq f(\mathbf{x}_t) - \gamma_t\langle \nabla f(\mathbf{x}_t)_{|E_s}, \mathbb{E}[\mathbf{g}_t^s]_{|E_s}\rangle + \frac{\gamma_t^2 L}{2}\mathbb{E}\left\|\mathbf{g}_{t|E_s}^s\right\|^2$$

$$= f(\mathbf{x}_t) - \gamma_t\langle \nabla f(\mathbf{x}_t)_{|E_s}, \mathbb{E}[\mathbf{g}_t^s]_{|E_s}\rangle + \frac{\gamma_t^2 L}{2}\mathbb{E}(\left\|\mathbf{g}_{t|E_s}^s - \mathbb{E}\mathbf{g}_{t|E_s}^s\right\|^2 + \left\|\mathbb{E}\mathbf{g}_{t|E_s}^s\right\|^2)$$

$$\leq f(\mathbf{x}_t) - \gamma_t\langle \nabla f(\mathbf{x}_t)_{|E_s}, \nabla f^s(\mathbf{x}_t^s)_{|E_s}\rangle + \frac{\gamma_t^2 L}{2}\mathbb{E}(\|\mathbf{g}^s - \mathbb{E}\mathbf{g}_t^s\|^2 + \left\|\nabla f^s(\mathbf{x}_t^s)_{|E_s}\right\|^2)$$

$$\leq f(\mathbf{x}_t) - \gamma_t\langle \nabla f(\mathbf{x}_t)_{|E_s}, \nabla f^s(\mathbf{x}_t^s)_{|E_s}\rangle + \frac{\gamma_t^2 L}{2}\left\|\nabla f^s(\mathbf{x}_t^s)_{|E_s}\right\|^2 + \frac{\gamma_t^2 L}{2}\sigma^2$$

$$\leq f(\mathbf{x}_t) - \gamma_t\alpha_t(1 - \frac{\gamma_t}{2\alpha_t}L)\left\|\nabla f^s(\mathbf{x}_t^s)_{|E_s}\right\|^2 + \frac{\gamma_t^2 L}{2}\sigma^2$$

$$\leq f(\mathbf{x}_t) - \frac{\gamma_t}{2}\alpha_t\left\|\nabla f^s(\mathbf{x}_t^s)_{|E_s}\right\|^2 + \frac{\gamma_t^2 L}{2}\sigma^2$$

$$\leq f(\mathbf{x}_t) - \frac{\gamma}{2}\alpha_t^2\left\|\nabla f^s(\mathbf{x}_t^s)_{|E_s}\right\|^2 + \frac{\gamma^2 L}{2}\sigma^2$$

Where in the last equation we used the facts that $\alpha_t \leq 1$ and $\gamma_t = \alpha_t\gamma$. $\square$

We now prove Theorem 1.

*Proof.* We first define $F_t := \mathbb{E}f(\mathbf{x}_t) - (\min_{\mathbf{x}} f(\mathbf{x}))$. By rearranging Lemma 2, we have

$$\frac{1}{2}\mathbb{E}\alpha_t^2\left\|\nabla f^s(\mathbf{x}_t^s)_{|E_s}\right\|^2 \leq \frac{F_t - F_{t+1}}{\gamma} + \frac{\gamma L}{2}\sigma^2. \tag{8}$$

Next, with telescoping summation, we have

$$\frac{1}{T}\sum_{t=0}^{T-1}\mathbb{E}\alpha_t^2\left\|\nabla f^s(\mathbf{x}_t^s)_{|E_s}\right\|^2 \leq \frac{2(F_0 - F_{T-1})}{T\gamma} + \gamma L\sigma^2 \leq \frac{2F_0}{T\gamma} + \gamma L\sigma^2 \tag{9}$$

We now can prove the first of part Theorem 1 by setting the step size $\gamma$ to be $\mathcal{O}(\min\{\frac{1}{L}, (\frac{F_0}{\sigma^2 T})^{\frac{1}{2}}\}$ as in (Mohtashami et al., 2021).

To prove the convergence of the model of interest (the second part),

$$\frac{1}{T}\sum_{t=0}^{T-1}\|\nabla f(\mathbf{x}_t)\|^2 = \frac{1}{T}\sum_{t=0}^{T-1}\frac{\|\nabla f(\mathbf{x}_t)\|^2}{\alpha_t^2\left\|\nabla f^s(\mathbf{x}_t^s)_{|E_s}\right\|^2}\alpha_t^2\left\|\nabla f^s(\mathbf{x}_t^s)_{|E_s}\right\|^2$$

$$= \frac{1}{T}\sum_{t=0}^{T-1}q_t^2\alpha_t^2\left\|\nabla f^s(\mathbf{x}_t^s)_{|E_s}\right\|^2 \leq q^2\frac{1}{T}\sum_{t=0}^{T-1}\alpha_t^2\left\|\nabla f^s(\mathbf{x}_t^s)_{|E_s}\right\|^2 \tag{10}$$

where

$$q_t = \frac{\|\nabla f(\mathbf{x}_t)\|}{\alpha_t\left\|\nabla f^s(\mathbf{x}_t^s)_{|E_s}\right\|} \quad \text{and} \quad q = \max_{t\in[T]} q_t. \tag{11}$$

Table 6: Parameters for federated experiments

| Dataset | #clients | #clients_per_epoch | batch_size | #epochs |
|---|---|---|---|---|
| EMNIST | 3400 | 68 | 20 | 1500 |
| CIFAR-10 | 100 | 10 | 50 | 2000 |
| CIFAR-100 | 500 | 40 | 20 | 3000 |
| BraTS | 10 | 10 | 3 | 100 |
| | #epoch_per_client | #stages ($S$) | $T_s$ | #epochs_for_warmup |
| EMNIST | 1 | 3 | 250 | 5 |
| CIFAR-10 | 5 | 4 | 250 | 0 |
| CIFAR-100 | 1 | 4 | 375 | 25 |
| BraTS | 3 | 4 | 25 | 0 |

By definition $q \geq q_t$ for all $t \in [T]$, we reach the last inequality and combine it with the first part of the theorem.

$$\frac{1}{T} \sum_{t=0}^{T-1} \alpha_t^2 \|\nabla f^s(\mathbf{x}_t)\|^2 \leq \frac{\epsilon}{q^2} \tag{12}$$

$\square$

Using $\frac{\epsilon}{q^2}$ as the new threshold, we immediately prove the second part.

## B    IMPLEMENTATION DETAILS

We describe details of the datasets used in Section 4 and present the hyper-parameters in Table 6.

**CIFAR-10.**   We conduct experiments on CIFAR-10 datasets for federated learning, following the setup of previous work (McMahan et al., 2017). The dataset is divided into 100 clients randomly, namely iid distributions for every client. We adopt the same CNN architecture with 122,570 parameters.

**CIFAR-100.**    We follow the federated learning benchmark of CIFAR-100 proposed in (Reddi et al., 2021) to conduct the experiments on CIFAR-100. We use ResNet-18/-152 (batch norm are replaced with group norm (Hsieh et al., 2020)) and VGG-16/-19 in the centralized setting, while only considering ResNet-18 in the federated experiments. This setup allows us to evaluate the federated learning systems on non-IID distributions, where we use the splits as suggested in (Reddi et al., 2021).

**EMNIST.**  We follow the benchmark setting in (Reddi et al., 2021) to experiment. There are 3,400 clients and 671,585 training examples distributed in a non-iid fashion. The models are eventually evaluated on 77,483 examples, resulting in a challenging task.

**BraTS.**  In addition to image classification, we conduct experiments on brain tumor segmentation based on (Sheller et al., 2020). We train a 3D-Unet on the BraTS2018 dataset, which includes 285 MRI scans annotated by five classes of labels. The network has 9,451,567 parameters. The training set is randomly partitioned into ten clients. All clients participate in every training round and locally train their models for three local epochs. This setting matches the practical medical applications. Institutions often own relatively stable network conditions, and the data are rare and of high resolution.

**Architectures.**    ConvNets for EMNIST and MNIST consist of two convolution layers, termed Conv1 and Conv2, followed by two fully connected layers, termed FC1 and FC2. To apply progressive learning with $S = 3$, we set Conv1, Conv2, FC1 to be the three stages, namely $E_i$, and FC2 to the final head, namely $G_S$. As for VGGs, we divide the whole networks into five components according to the max-pooling layers. We combine the first two to be $E_1$ and set the others to be the remaining $E_i$ under the setting $S = 4$. To apply ProgFed to ResNets He et al. (2016), we first replace the batch normalization layers with group normalization. By convention, ResNets have

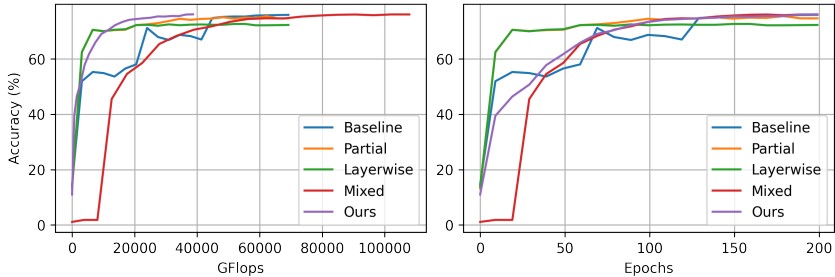

Figure 8: Performance vs. computation costs and Performance vs. epochs when comparing our method to different updating strategies.

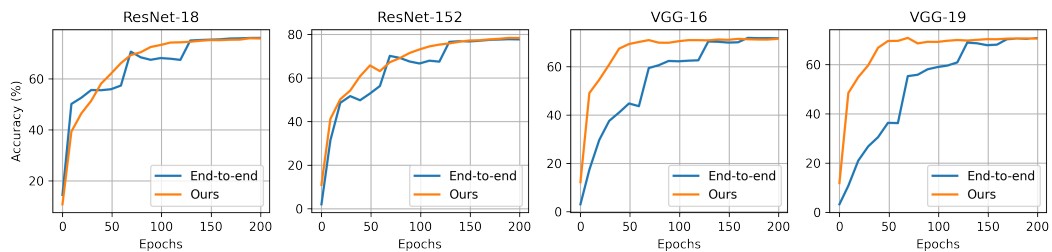

Figure 9: Accuracy (%) vs. Epochs on CIFAR-100 in the centralized setting.

five convolution components, i.e. Conv1, Conv2_x, Conv3_x, Conv4_x, and Conv5_x. We combine Conv1 and Conv2_x to be $E_1$ and all the other components to be the remaining $E_i$. It thus matches $S = 4$ in our setting.

## C   MORE RESULTS

### C.1   COMPARISON BETWEEN UPDATE STRATEGIES

As described in Section 4.5, we compare ProgFed to other baselines. We additionally report the performance vs. computation costs and performance vs. epochs in Figure 8, where `Ours` reaches comparable performance while consuming the least cost.

### C.2   COMPUTATION EFFICIENCY

We present more experiments in the centralized setting to prove the computation efficiency of our method. Figure 9 presents accuracy vs. epochs with four architectures on CIFAR-100. The result

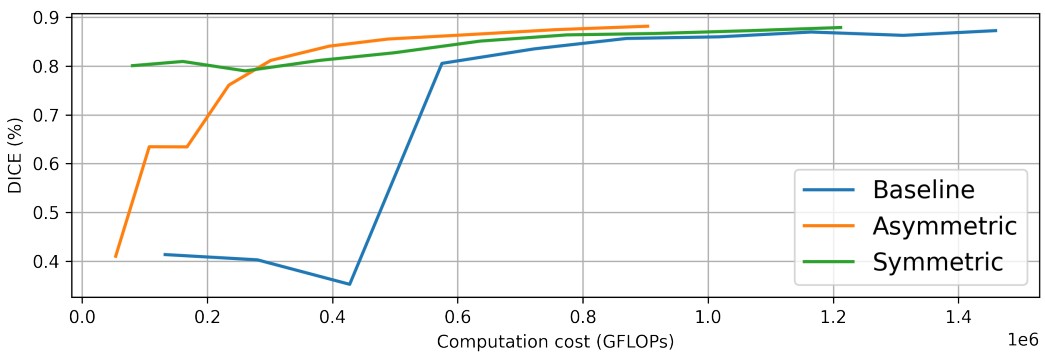

Figure 10: DICE (%) vs. computation costs on BraTS.

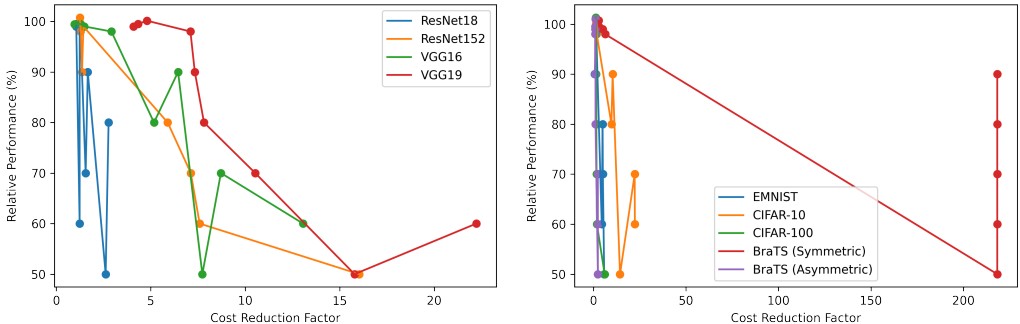

Figure 11: Computation acceleration at different percentage of performance. The orange bar indicates the best performance of our method.

Figure 12: Computation cost reduction at {50%, 60%, 70%, 80%, 90% 98%, 99%, 99.95%, *best*} of the baseline performance in the centralized setting.

Figure 13: Communication cost reduction at {50%, 60%, 70%, 80%, 90% 98%, 99%, 99.95%, *best*} of the baseline performance in the federated setting.

indicates that our method converges comparably faster to end-to-end training in practice. Figure 11 presents Figure 3 in bar charts. Similar to Figure 3, our method improves across architectures while VGGs benefit even more from our method. Figure 10 presents the computation costs of 3D-Unets on the BraTS dataset. We make the first observation that tumor segmentation requires heavy computation. Interestingly, even though the earlier stages of *Symmetric* consume much fewer communication costs (Figure 5), they require more computation costs than *Asymmetric*. It might root from the higher resolution of feature maps that *Symmetric* keeps and thus lead to a trade-off between communication and computation costs.

Figure 11 extend Figure 3 to a larger range {50%, 60%, 70%, 80%, 90% 98%, 99%, 99.95%, *best*}. The result shows that our method benefits across models and is especially efficient when training budgets are limited.

## C.3 COMMUNICATION EFFICIENCY

We present more experiments in the federated setting to prove the communication efficiency of our method. To complement Figure 5, we additionally visualize performance vs. communication costs and performance vs. epochs in Figure 14. Although our method causes performance fluctuation in some datasets, the performance recovers very quickly. Figure 15 presents Figure 4 in bar charts. The results show that our method saves considerable costs in almost all settings except for EMNIST at 95%. It is because both baseline and our method improve fast at the beginning while our method stands out in the latter phase of training (e.g. after 98%). The result also supports that our method improves across datasets.

We present Figure 4 in a region that models are applicable. Here, we plot the figure with a larger range {50%, 60%, 70%, 80%, 90% 98%, 99%, 99.95%, *best*} in Figure 13. The result is consistent, showing that our method benefits across datasets, and is efficient when granted limited training budgets.

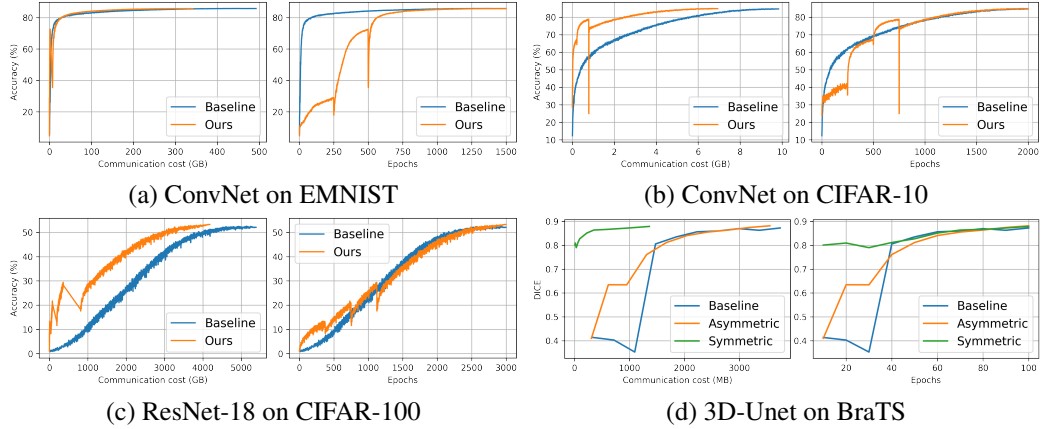

(a) ConvNet on EMNIST          (b) ConvNet on CIFAR-10

(c) ResNet-18 on CIFAR-100          (d) 3D-Unet on BraTS

Figure 14: Accuracy vs. computation costs and accuracy vs. epochs in the federated setting. (a)(b)(c) shows the result for three classification tasks; (d) shows the result for the segmentation task, where two update strategies *Symmetric* and *Asymmetric* are adopted for 3D-Unet.

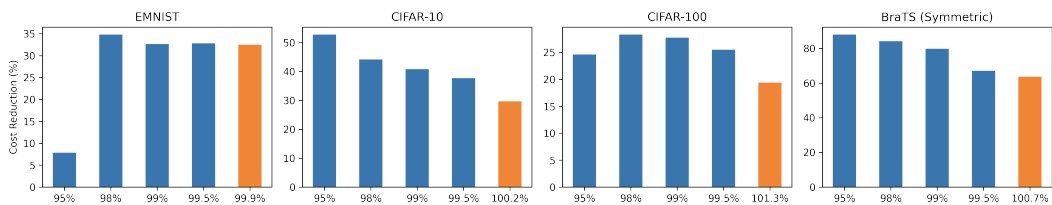

Figure 15: Communication cost reduction at different percentage of performance. The orange bar indicates the best performance of our method.

## C.4 VISUALIZATION OF FEDERATED SEGMENTATION

We visualize the outputs of 3D-Unet (see Section 4.3) in Figure 16. The result matches the number reported in Table 3 that the models perform similarly after they have converged. However, the communication cost consumption is disparate. *Symmetric* only consumes $36.40\%$ while the others either do not save any cost or marginally improve it. We visualize the results in Figure 17 when granted limited communication budgets. Note that even with the same cost, *Symmetric* and *Asymmetric* may stay in different stages because *Symmetric* starts from the outer part, consisting of significantly fewer parameters. We find that Baseline and *Asymmetric* fail to compress the models with $0.18\%$ since their models are of size 34 MB (i.e. $0.908\%$), while only *Symmetric* achieve it. Interestingly, *Symmetric* has produced promising results at $0.18\%$ of costs. It suggests that our method significantly facilitates learning even given limited communication budgets. Meanwhile, we hope these findings could inspire more further work on medical learning problems.

Table 7: Raw results on CIFAR-100 with four architectures in the centralized setting (to complement Table 2).

|  | End-to-End | | | | | ProgFed (Ours) | | | | |
|---|---|---|---|---|---|---|---|---|---|---|
|  | Seed1 | Seed2 | Seed3 | Mean | Std | Seed1 | Seed2 | Seed3 | Mean | Std |
| ResNet-18 | 75.95 | 76.12 | 76.17 | 76.08 | 0.12 | 75.57 | 76.13 | 75.83 | 75.84 | 0.28 |
| ResNet-152 | 77.69 | 77.44 | 78.19 | 77.77 | 0.38 | 78.95 | 78.46 | 78.31 | 78.57 | 0.33 |
| VGG16 (bn) | 71.94 | 71.77 | 71.65 | 71.79 | 0.15 | 71.36 | 72.05 | 71.21 | 71.54 | 0.45 |
| VGG19 (bn) | 69.47 | 71.25 | 71.70 | 70.81 | 1.18 | 71.30 | 70.45 | 70.95 | 70.90 | 0.43 |

Table 8: Raw results in federated settings on EMNIST, CIFAR-10, and CIFAR-100 (to complement Table 3).

| | Seed1 | Seed2 | Seed3 | Mean | Std |
|---|---|---|---|---|---|
| **EMNIST** | | | | | |
| Baseline | 85.63 | 85.77 | 85.85 | 85.75 | 0.11 |
| Ours | 85.65 | 85.62 | 85.73 | 85.67 | 0.06 |
| **CIFAR-10** | | | | | |
| Baseline | 84.62 | 84.82 | 84.56 | 84.67 | 0.14 |
| Ours | 84.64 | 84.73 | 85.19 | 84.85 | 0.30 |
| **CIFAR-100** | | | | | |
| Baseline | 51.79 | 51.86 | 52.58 | 52.08 | 0.44 |
| Ours | 53.33 | 53.16 | 53.21 | 53.23 | 0.09 |

Table 9: Raw results on BraTS in the federated setting (to complement Table 3).

| | Seed 1 | Seed 2 | Seed 3 | Mean | Std |
|---|---|---|---|---|---|
| Baseline | 87.29 | 86.51 | 86.51 | 86.77 | 0.45 |
| Idea1 (Ours) | 88.19 | 87.22 | 87.57 | 87.66 | 0.49 |
| Idea2 (Ours) | 87.92 | 87.98 | 87.98 | 87.96 | 0.03 |

Table 10: Federated ResNet-18 on CIFAR-100 with compression. LQ-X denotes linear quantization followed by used bits representing gradients, and SP-X denotes sparsification followed by the percentage of kept gradients. (to complement Table 4).

| | Float | LQ-8 | LQ-4 | LQ-2 |
|---|---|---|---|---|
| *Accuracy (%)* | | | | |
| Baseline | $52.54 \pm 0.44$ | $49.40 \pm 0.75$ | $49.55 \pm 0.59$ | $47.26 \pm 0.29$ |
| Ours | $\mathbf{53.23 \pm 0.09}$ | $\mathbf{53.07 \pm 1.00}$ | $\mathbf{52.32 \pm 0.15}$ | $\mathbf{52.87 \pm 0.54}$ |
| *Compression ratio (%)* | | | | |
| Baseline | 100 | 25.00 | 12.50 | 6.25 |
| Ours | **77.10** | **19.28** | **9.64** | **4.82** |
| | SP-25 | SP-10 | LQ-8 +SP-25 | LQ-8 +SP-10 |
| *Accuracy (%)* | | | | |
| Baseline | $51.23 \pm 0.56$ | $51.79 \pm 0.10$ | $49.67 \pm 1.58$ | $50.25 \pm 1.03$ |
| Ours | $\mathbf{52.00 \pm 0.19}$ | $\mathbf{51.86 \pm 0.23}$ | $\mathbf{52.19 \pm 0.03}$ | $\mathbf{52.24 \pm 0.12}$ |
| *Compression ratio (%)* | | | | |
| Baseline | 25.00 | 10.00 | 6.25 | 2.50 |
| Ours | **19.28** | **7.71** | **4.82** | **1.93** |

Table 11: Results with FedAvg and FedProx on EMNIST and CIFAR-100.

|  | EMNIST | CIFAR-100 |
| --- | --- | --- |
| End-to-end+FedAvg | 85.75 | 52.54 |
| ProgFed+FedAvg | 85.67 | 53.36 |
| End-to-end+FedProx | 86.36 | 53.25 |
| ProgFed+FedProx | 86.03 | 52.30 |

### C.5 RAW NUMBERS AND MORE STATISTICS

Table 7 8, and 9 present the raw numbers of the experiments in Table 2 and 3 over three random seeds. Table 10 presents the standard deviations of Table 4 over three random seeds.

### C.6 GENERALIZABILITY OF PROGFED BEYOND FEDAVG

We show that ProgFed can generalize to other federated optimizations beyond FedAvg. In this section we show that ProgFed can generalize to advanced optimizations. We combine our method with FedProx and conduct experiments on EMNIST and CIFAR-100 with ConvNets and ResNet-18, respectively. Table 11 shows that both end-to-end training and ProgFed improves over FedAvg on EMNIST when applying FedProx. On the other, our method with FedProx shows little improvement on CIFAR-100 while end-to-end training benefits from FedProx (52.54 vs. 53.25). However, despite the improvement from FedProx, it remains comparable to ProgFed with FedAvg.

## D MORE RELATED WORK

We discuss more related work in this section.

**Progressive Learning.** The core idea of progressive learning (PL) is to train the model from easier tasks (e.g., low-resolution outputs or shallower models) to difficult but desired tasks (e.g., high-resolution outputs or deeper models). It was originally proposed to stabilize the training process and has been widely considered in vision tasks such as image synthesis (Karras et al., 2018), image super-resolution (Wang et al., 2018), facial attribute editing (Wu et al., 2020) and representation learning (Li et al., 2019). One of the most representative methods is PGGAN (Karras et al., 2018), which trains GANs first for generating 4x4 images with a shallow network and progressively extends the network for generating images of size 1024x1024. In addition to stabilizing the training process, PL naturally reduces the computation demands since it typically employs smaller models before it reaches the full model. This even benefits federated learning since the message size also decreases. Despite the benefits, little work has investigated applying PL to federated learning. We note that although some works (He et al., 2021; Belilovsky et al., 2020) also consider partially training, they often seek solutions to fixing layers rather than considering training difficulty, and the application in general federated tasks remain unexplored.

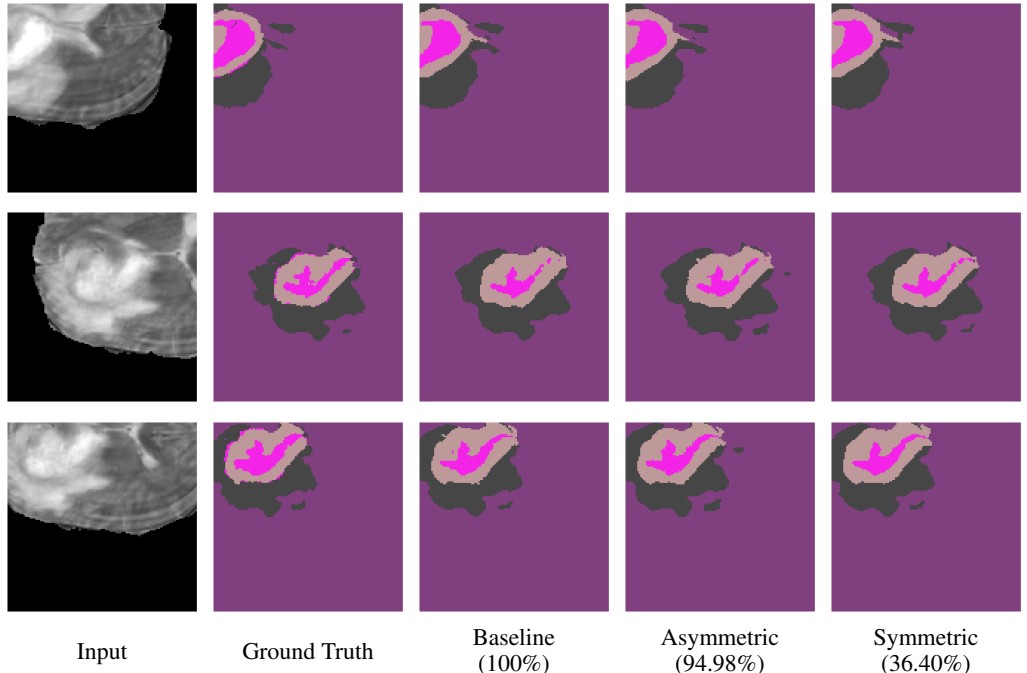

| Input | Ground Truth | Baseline (100%) | Asymmetric (94.98%) | Symmetric (36.40%) |

Figure 16: Visualization of federated segmentation. From left to right: Input, Ground Truth, Baseline, Asymmetric, and Symmetric updating strategies. Despite the comparable performance, *Symmetric* consumes significantly fewer communication costs.

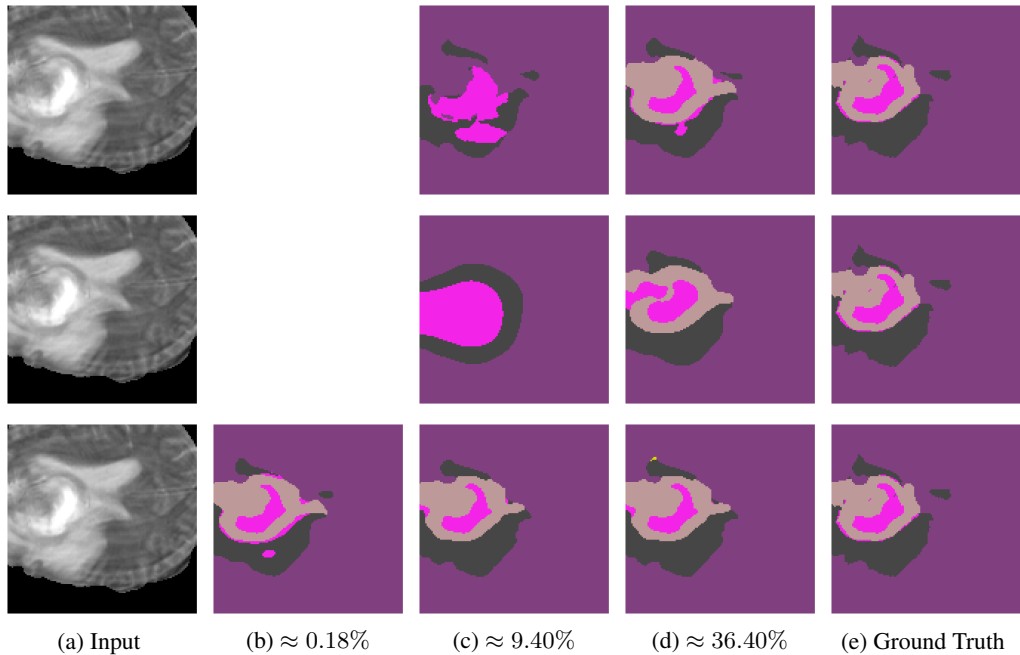

| (a) Input | (b) ≈ 0.18% | (c) ≈ 9.40% | (d) ≈ 36.40% | (e) Ground Truth |

Figure 17: Segmentation results under {≈ 0.18%, ≈ 9.40% , ≈ 36.40%} of communication costs of the converged baseline. From top to bottom: baseline, *Asymmetric* (Ours), and *Symmetric* (Ours). Only *Symmetric* can achieve 0.18% (6.7536 MB) compression ratio, since the size of the other models is already around 34 MB (i.e. 0.908%).

