# OpenReview forum: "ProgFed: Effective, Communication, and Computation Efficient Federated Learning by Progressive Training"
_ICLR.cc/2022/Conference — ICLR 2022 Submitted_

### Official Review · Reviewer_3zVc · 2021-10-27

**Correctness:** 3
**Technical Novelty And Significance:** 2
**Empirical Novelty And Significance:** 3
**Recommendation:** 5
**Confidence:** 3

**Main Review:**

Please see "Summary Of The Paper" for the overview of the paper.

The paper introduces "ProgFed", which is an algorithm designed for federated learning scenarios. The authors use theoretical and empirical results trying the show the superiority of ProgFed over other federated learning methods when applying to neural networks. However, I think that there are several weaknesses in this paper.

1. The first weakness (major weakness) comes from the  "algorithm" part. If I understand correctly, the novelty of ProgFed is applying the progressive training technique into the federated learning setting. However, progressive training is an existed method trying to speed up the neural network training in the single machine setting. In the single machine case, it is easy to apply progressive training techniques with different optimizers, e.g. RMSProp, ADAM, SGD, or other optimization algorithms. In the federated learning setting, it is also natural to apply progressive training with other federated optimization algorithms, e.g. local methods like FedAvg (Local-SGD), SCAFFOLD, FedPAGE, or compression schemes like quantization, Sign-SGD, etc. For ProgFed,  if I understand correctly, it just applies the progressive training technique with the Local-SGD algorithm. Thus the first weakness comes from the algorithmic novelty perspective.

2. The second weakness (major weakness) comes from the model and the theoretical analysis. The data heterogeneity is the most important aspect that differentiates federated optimization from distributed optimization. If the data on different clients are not iid, then the gradients on different clients are not the same. In ProgFed algorithm (Algorithm 1) Line 16, the algorithm uses local update $x_{c,j+1}^s = x_{c,j}^s - \eta g_c^s(x_{c,j}^s)$ for stage $s$, client $c$, local step $j$. Here, $\nabla f_c(\cdot)$ should be different in the federated learning setting. However, in the assumption part and the proofs, I do not see any assumptions to control the data heterogeneity (like $(G,B)$-BGD). Based on the fact that FedAvg and Local-SGD need assumptions like $(G,B)$-BGD to guarantee the convergence, I believe that either the model does not consider data heterogeneity (which is a main drawback for the federated learning applications), or the proof needs to modify in order to support data heterogeneity.

3. The third weakness (minor weakness) comes from the empirical part. For the experiments, the authors compare ProgFed with very straightforward optimization (end-to-end) and several compression schemes. However, as I stated previously, ProgFed actually consists of two modules, progressive training and Local-SGD, which behave at a different level. Directly comparing ProgFed with end-to-end training does not show the effectiveness of ProgFed, I suggest the authors add more experiments, such as ProgFed vs Progressive training + simple SGD, to demonstrate the effectiveness of local-steps with the existence of progressive training, or ProgFed vs Progressive training + quantization, to demonstrate whether the progressive training combines better with local steps methods or compression methods, or ProgFed VS Local-SGD, to show the power of progressive training. Besides, I think the data split for different datasets can be moved to the main content.

Please point out if my understanding is wrong, and I will consider raising my score if my concerns and questions are addressed or answered.

**Summary Of The Paper:**

The paper introduces "ProgFed", which is an algorithm designed for federated learning scenarios.

ProgFed applies the idea of progressive training. Instead of training the neural network from end to end, it first trains the shallow layers, and then gradually train the deep layers, hoping that: 1) training the shallow layers first will learn some simple and meaningful features which will help training the deep layers, 2) and thus reduces the communication cost for the federated learning applications. Besides, the ProgFed algorithm also applies the local steps (Local-SGD or FedAvg) in order to decrease the communication cost during the model training. The ProgFed algorithm works not only for the feed-forward neural nets but also the U-nets, which consist of an encoder and a decoder.

For the theoretical part, the authors show that ProgFed has similar convergence results compared with SGD.

For the empirical (experimental) part, the authors use experiments trying to show the superiority of ProgFed over other existed federated learning methods, e.g. the trivial end-to-end training procedure, the communication compression methods (e.g. quantization)

**Summary Of The Review:**

In summary, I have the following concerns:

1. The proposed ProgFed algorithm seems a little bit incremental, since it seems like an "A+B" method where A is progressive training technique and B is FedAvg (local-SGD).
2. For the model or the theoretical analysis, either the model does not consider data heterogeneity or the theoretical analysis needs to be modified.
3. (minor concern) ProgFed is a combination of two techniques, and comparing with compression schemes like quantization is not very fair.

---

> ### Author Response · Authors · 2021-11-20
> **Reply to Reviewer 3zVc: clarifying novelty and experiment settings**
>
> We appreciate Reviewer 3zVc's comments. We would like to recap a few points of our work: (1) Although progressive learning has existed, little work has studied its application in federated learning. We make the first attempt to analyze and apply it to general federated tasks. (2) We show that ProgFed can work well in non-iid settings and potentially work with advanced optimization methods. We now address the comments below.
>
> > Compatibility with other federated optimization methods.
>
> We additionally conduct experiments with FedProx on EMNIST and CIFAR-100, respectively. On EMNIST, both the baseline (86.36) and our method (86.03) improve the ones with FedAvg. On CIFAR-100, the end-to-end+FedProx improves (53.36) over FedAvg (52.54). On the other hand, our method with FedProx does not improve (53.25 vs. 52.3) while remaining comparable to FedAvg. These results indicate that our method is compatible with optimizations beyond FedAvg.
>
> > The proof and the model do not consider data heterogeneity.
>
> We have shown in experiments (Table 3, 4 and Figure 4, 5, 6) that our method is amenable to non-i.i.d settings in several challenging benchmarks, including EMNIST (3400 clients, non-iid) and CIFAR-100 (500 clients, non-iid). Despite no data heterogeneity assumption in Theorem 1, we mathematically analyze the convergence rate of progressive learning and demonstrate the possibility to adapt progressive learning to federated learning. We believe this could shed light on a new way to reduce training resource demands and benefit the community, especially the practitioners.
>
> > The authors compare ProgFed with very straightforward optimization (end-to-end) and several compression schemes. However, as I stated previously, ProgFed actually consists of two modules, progressive training and Local-SGD, which behave at a different level. Directly comparing ProgFed with end-to-end training does not show the effectiveness of ProgFed
>
> The major contribution of ProgFed is to introduce progressive learning to federated learning, which is orthogonal to optimizers. For instance, one could form ProgFed+FedAvg or ProgFed+FedProx. The counterparts in most of the experiments are end-to-end+FedAvg vs. progressive learning (ProgFed)+FedAvg. They **fairly** outline the contribution of ProgFed. Similarly, the main idea of Table 4 is to show that our method is compatible with traditional compression techniques rather than outperforms them. The experiment verifies the compatibility and shows the potential to achieve higher compression ratios.
>
> > Editorial comments
>
> We add a few sentences in Sec. 4.1 to describe the data splits.

---

> > ### Comment · Reviewer_3zVc · 2021-11-27
> > **After Rebuttal**
> >
> > Dear Authors,
> >
> > Thanks for the reply. I changed my score since the reply alleviates some of my concerns, but my main concerns still exist.
> >
> > From a novelty point of view, it is a drawback since it just applies progressive training into federated learning without introducing many new ideas. However, there must be some papers doing this work.
> >
> > From a theoretical point of view, I believe the theory is not enough. The proof does not introduce new messages to me and it is just some basic manipulations of the inequalities. The major concern for the theoretical part is still the assumption that the data on each client is homogeneous, which is not what happens in federated learning or may greatly limit the applications in federated learning. The success of empirical studies with heterogeneous data cannot improve the theoretical contribution.
> >
> > From the empirical perspective, I think this paper's empirical study is pretty well. However, I think it is not enough for the current theory. If this paper has a better theory, then the experiments are pretty well. However, without strong theoretical contributions, I think the author still needs to improve the experiments, i.e., adding progressive training techniques with other federated learning methods like SCAFFOLD or some communication compression techniques, e.g. QSGD, DIANA etc.
> >
> > I have the following suggestions for the authors for their camera-ready version (if lucky) or the future version: First focus on the theoretical contribution or the empirical contribution. If the main contribution is theoretical, then try to directly show the convergence results without assuming the data are homogeneous. Using assumptions like $(G,B)$-BGD (bounded gradient dissimilarity) or some similar assumptions are pretty OK. If the main contribution is empirical, I strongly suggest the authors present progressive training as an orthogonal method to the optimization algorithms, and test with much more algorithms like SCAFFOLD, QSGD etc.

---

> > > ### Author Response · Authors · 2021-11-29
> > > **Clarifying novelty, details and provide additional experiments**
> > >
> > > We appreciate the comments and thank Reviewer 3zVc for reconsidering the score. We address the comments as follows and hope our reply can address the reviewer's concerns. In short, we had already applied QSGD in our paper and show that our method can extend to FedAdam, which is one of the most popular optimization methods.
> > >
> > > > From a novelty point of view, it is a drawback since it just applies progressive training into federated learning without introducing many new ideas. However, there must be some papers doing this work.
> > >
> > > To our knowledge, there is no work studying this simple but effective method in generic federated learning. We would like to know the references if the reviewer is aware of any.
> > >
> > > > The major concern for the theoretical part is still the assumption that the data on each client is homogeneous, which is not what happens in federated learning or may greatly limit the applications in federated learning.
> > >
> > > We agree that theorems with the assumption might provide more insights. However, we have shown in our experiments that our method does not limit the applications. It can generalize to various architectures (ConvNets, ResNets, U-nets), tasks (image classification and 3D-volume segmentation), optimizations (including FedAvg, FedProx, FedAdam), and non-iid data distribution (We follow the benchmark [1] for Non-IID data splits in EMNIST and CIFAR-100)
> > >
> > > > I think the author still needs to improve the experiments, i.e., adding progressive training techniques with other federated learning methods like SCAFFOLD or some communication compression techniques, e.g. QSGD, DIANA etc.
> > >
> > > As per Reviewer 1N8v, we additionally extend our method to FedAdam, which is also one of the most common federated optimization methods [1]. In contrast to FedProx, which considers additional regularization on clients, FedAdam considers global momentums for server updating. It is observed that both the baseline (56.21) and our method (60.55) significantly outperform the ones with FedAvg (52.54 and 53.23, respectively). Overall, the experiment has demonstrated the potential to adapt to other optimization methods.
> > >
> > > Regarding compression techniques, in fact, we have already adopted QSGD in our paper (Table 4). The compression method [2] applied/cited in our paper is an improved version of QSGD. Similar to QSGD, it also adopts probabilistic quantization but additionally introduces Hadamard rotation to stabilize linear quantization. We will cite QSGD [4] in the next version to avoid confusion.
> > >
> > > [1] Reddi, Sashank J., et al. "Adaptive Federated Optimization." International Conference on Learning Representations. 2021.
> > >
> > > [2] Konečný, Jakub, et al. "Federated learning: Strategies for improving communication efficiency." arXiv preprint arXiv:1610.05492 (2016).
> > >
> > > [3] Suresh, Ananda Theertha, et al. "Distributed mean estimation with limited communication." International Conference on Machine Learning. PMLR, 2017.
> > >
> > > [4] Alistarh, Dan, et al. "QSGD: Communication-efficient SGD via gradient quantization and encoding." Advances in Neural Information Processing Systems 30 (2017): 1709-1720.

---

### Official Review · Reviewer_5a4S · 2021-10-29

**Correctness:** 4
**Technical Novelty And Significance:** 3
**Empirical Novelty And Significance:** 3
**Recommendation:** 8
**Confidence:** 4

**Main Review:**

This submission is good; the paper is well-written, the idea is simple and seems to work well in practice. Furthermore, the experimental evaluation is thorough and considers multiple datasets and architectures. There is also a nice bonus that the proposed method is orthogonal to more traditional approaches that target communication cost reduction. For this reason, I am leaning towards recommending acceptance.

Having said that, there are some things that I would like some input from the authors:
- It seems that currently you throw away the auxiliary heads obtained during the progressive training phase. This seems like a waste; why not reuse these heads for inference? For example, one could argue about some “voting” between the heads or even “early exiting” of datapoints if an early head is confident enough about the prediction (this would reduce the computational complexity at inference time as well).  Does the performance of each head become worse after the end-to-end fine-tuning?
- While the provided proof is welcome, unless I am missing something, it seems to be in the centralised setting. Can it be extended to the federated setting? As the main motivation of this work is the federated setting, it makes more sense to prove convergence in such scenarios.
- In the experiments you report the mean after three experiment runs. Why not show the standard deviation / error as well? This will allow one to better judge the significance of the results. Currently it is a bit unclear.
- In Figure 2 it is weird that the end-to-end training has a bit of instability that is not present in progressive training. Could you elaborate on why this is the case? Furthermore, at Figure 5 ProgFed seems to have quite some aggressive dips in performance (I guess when switching stages). This is especially pronounced at CIFAR 10. Could the authors expand upon this and whether such instabilities can be hindering in practice?
- Do you have any indication or intuition as to why progressive training helps with quantization / compression?
- For section 4.5; does the “Layerwise” strategy use only one head, i.e., the one at the end? If yes, why not use a separate head (i.e, such as ProgFed)? How does Layerwise + end-to-end fine-tuning perform compared to ProgFed? Furthermore, the difference between ProgFed and “Random is small”, so some standard error indication would help.  Finally, the “Random” strategy is missing from Figure 8 and it would be great if you could elaborate on what is the extra memory that “Random” needs.

As for other minor things:
- The plots such as the one at Figure 3 are kind of confusing and not very clear. I would suggest that the authors convert them to bar plots (such as the ones they have in the appendix) instead.
- There is a reference that is listed twice (the “Sparsified SGD with memory”)
- “Layerwise” referred to as “Laywise”


**Summary Of The Paper:**

The issue that this work attempts to solve is that of computation and communication efficiency in federated learning. This is realised via ProgFed, a method that progressively trains neural network blocks of layers in $S$ “stages”. For the first $S - 1$ stages only a subset of layers is selected to be optimised, with the help of an optional (depending on the architecture) additional head to provide the training signal (which is discarded after that particular stage is finished). At the last stage, the authors propose to fine-tune the entire architecture in an end-to-end manner. Due to the specific nature of this form of training, the first $S-1$ stages of training require less compute (due to not needing to perform the forward / backward passes on the layers not selected) and also less communication (as the clients and server only need to communicate the specific set of layers that is optimised). The authors prove that such a training procedure converges in a rate similar to the one of standard end-to-end training and demonstrate in extensive experiments the benefits of ProgFed. As an extra bonus, the authors also show that ProgFed is more amenable to compression / quantization of the federated messages, thus showing that the benefits of these, more traditional, approaches are additive to the ones of ProgFed.

**Summary Of The Review:**

Good submission with a simple idea that seems to work well in practice. Extensive evaluation and nice compatibility with more traditional compression and quantization. There are some points that I would like some input from the authors which, if turned out positive, would strengthen this work further. For this reason, I am leaning towards acceptance.

---

> ### Author Response · Authors · 2021-11-20
> **Reply to Reviewer 5a4S: clarifying and discussing implementation details**
>
> We thank Reviewer 5a4S for the constructive comments and interesting discussion. We address the comments below, and more experiments will follow soon.
>
> >   Why not reuse these heads for inference?
>
> Our goal is to deploy the same architecture as the end-to-end training one, but early-exit or voting scenarios might be possible future directions.
>
> > Does the performance of each head become worse after the end-to-end fine-tuning?
>
> The exact numbers will follow soon.
>
> >  Can the proof/Theorem 1 be extended to the federated setting?
>
> Yes under assumptions such as limited data heterogeneity.
>
> >  Standard deviation of the report numbers.
>
> We have added the standard deviation to the main paper.
>
> > The end-to-end training has a bit of instability that is not present in progressive training in Figure 2.
>
> The instability of the baselines comes from the learning scheduling, which is a stepwise function, and overfitting might cause slight instability. On the other hand, our method employs learning restarts, as mentioned in the implementation details in Sec 4.1. We also conducted experiments on baselines with a learning restart. We found that the final performance is similar. To adhere to the settings in prior work (DeVries & Taylor, 2017), we report the one with stepwise learning scheduling.
>
> > ProgFed seems to have quite some aggressive dips in performance in Figure 5. This is especially pronounced at CIFAR 10. Could the authors expand upon this and whether such instabilities can be hindering in practice?
>
> We found that the model takes some time to recover its performance after switching to a new stage. This phenomenon is particularly obvious in federated learning. The potential reason might be data heterogeneity. We also found that warm-up for the newly-added layers can improve the problem (53.23 vs. 51.09 on CIFAR-100 in federated learning), as discussed in the practical considerations in Sec 3.1. In practice, the performance recovers quickly. Another workaround might be to use a snapshot of the previous stage until the performance recovers.
>
> > Do you have any indication or intuition as to why progressive training helps with quantization/compression?
>
> Our intuition is that smaller models might suffer less from the quantization errors from the perspective of the total amount of noise. The extended models could learn upon the features, which leads to a more stable learning process.
>
> > Does the “Layerwise” strategy use only one head, i.e., the one at the end? If yes, why not use a separate head (i.e, such as ProgFed)?
>
> Yes. We assume the final goal is to train exactly the same model without introducing additional computation or memory overhead.
>
> > How does Layerwise + end-to-end fine-tuning perform compared to ProgFed?
>
> The experiment will follow up soon.
>
> > The difference between ProgFed and “Random is small”, so some standard error indication would help.
>
> We re-run the experiment in Table 5 over three random seeds to get the standard deviation. The numbers over 3 seeds are 73.78, 73.86, 75.5, leading to mean=74.38 and std=0.97.
>
> > The “Random” strategy is missing from Figure 8 and it would be great if you could elaborate on what is the extra memory that “Random” needs.
>
> We will add it to Figure 8. The additional cost includes indexing and momentum storing. That is, one needs to keep the momentum for all stages. We empirically observe that the momentum from previous stages could interfere with the performance. For instance, the performance degrades from 53.25% to 51.43% in a federated setting on CIFAR-100 with ResNet18 when reusing the momentum from previous stages. It suggests that the momentums might not be sharable among stages and might make the random strategy unfavorable since it requires additional memory caching on the clients.

---

> > ### Author Response · Authors · 2021-11-22
> > **Clarifying the performance of each head after fine-tuning**
> >
> > Dear Reviewer 5a4S,
> >
> > We would like to update you on the following comment.
> >
> > > Does the performance of each head become worse after the end-to-end fine-tuning?
> >
> > Yes, the performance of four heads of federated ResNet-18 on CIFAR-100 is 1.52%, 1.83%, 5.76%, and 53.23%. It might indicate that the learned features significantly change during fine-tuning.

---

> > > ### Comment · Reviewer_5a4S · 2021-11-25
> > > **Response to rebuttal**
> > >
> > > I would like to thank the authors for responding and addressing my comments. Based on their response, I have decided to increase my score. As a final comment, it would be good if the authors consider the following
> > >
> > > - Figures 3,4 are still confusing, would be good to replace them with bar chars.
> > > - "Layerwise + finetuning" result is still missing from Table 5.
> > > - The "Random" baseline is still missing from Figure 8.
> > > - Extend the proof to precisely accommodate the federated setting. Since the paper directly targets federated learning, it makes sense to do so.

---

> > > > ### Author Response · Authors · 2021-11-29
> > > > **Updates on fine-tuning layer-wise training**
> > > >
> > > > We thank Reviewer 5a4S for the suggestions and for raising the score. We will add the missing part in the next version. In the end, we would like to report the fine-tuning result on layerwise training. We fine-tune the trained model for another 200 epochs with lr=1e-3 and 1e-4, respectively: 73.05 (lr=1e-3) and 72.75 (lr=1e-4). The fine-tuning process only grants limited improvement, indicating that greedy training might hurt the performance. This finding seems to align with the paper [1].
> > > >
> > > > [1] Wang, Yulin, et al. "Revisiting Locally Supervised Learning: an Alternative to End-to-end Training." International Conference on Learning Representations. 2020.

---

### Official Review · Reviewer_1N8v · 2021-10-31

**Correctness:** 2
**Technical Novelty And Significance:** 3
**Empirical Novelty And Significance:** 3
**Recommendation:** 5
**Confidence:** 5

**Main Review:**

It is an interesting work on accelerating the training process of federated learning.  The reviewer still has several concerns about the current submission:
(1) It seems that the main benefit of progressive training for federated learning comes from the early stage during training.  Hence, we recommend the authors do more ablation studies on the hyperparameter $S$.
(2) For different stages,  how do the authors set the hyperparameters, such as learning rate.  Since in different stages, the depth of subnetworks is differents,  the hyperparameter tuning techniques may highly influence the training speed and final performance.
(3) Theorem 1 provides the convergence of the proposed ProgFed.  However, the convergence results do not indicate the benefit of progressive training for federated learning.
(4)  The experiments should be strengthened.  The main optimizer in ProgFed is fedavg.  The authors should incorporate the proposed ProgFed with several other types of local opitmizers, such as fedprox, scaffold, etc., to further show progressive training can be a versatile and strong technique for accelerating federated learning.

**Summary Of The Paper:**

The authors adopt the progressive training technique to accelerate the training process federated learning. The authors also establish the convergence rate of the proposed algorithm. Extensive experiments demonstrate the efficacy of the proposed algorithm.

**Summary Of The Review:**

see the comment above.

---

> ### Author Response · Authors · 2021-11-20
> **Reply to Reviewer 1N8v: Clarifying the compatibility and implementation details**
>
> We thank Reviewer 1N8v for the comments and address the concerns below.
>
> > More ablation studies on the hyperparameter S.
>
> We conducted an ablation study on Federated ResNet18 on CIFAR-100 with S={2, 3, 4, 5, 8}. The performance is 49.4 (S=2), 51.33 (S=3), 53.25 (S=4), 51.53 (S=5), and 50.07 (S=8), respectively. We observe that more stages lead to fewer communication costs while slowing the convergence, and adding too many parameters at a time (e.g. S=2) may hurt performance.
>
> > For different stages, how do the authors set the hyperparameters, such as learning rate?
>
> In the centralized setting, we apply learning rate restarts with a period of 10 epochs. The largest learning decays from 0.1 to 0 in every period. In the federated setting, both the baseline and our method apply the same learning rate scheduling, i.e. the only difference between the baseline and our method is progressive learning.
>
> > Theorem 1 provides the convergence of the proposed ProgFed. However, the convergence results do not indicate the benefit of progressive training for federated learning.
>
> The goal of the proof is to show that our method has a similar rate as standard training while per-iteration costs are lower than standard end-to-end training, demonstrating another way to reduce the resource demands in federated learning.
>
> > Compatibility of ProgFed with other optimization methods.
>
> We additionally conduct experiments with FedProx on EMNIST and CIFAR-100, respectively. On EMNIST, both the baseline (86.36) and our method (86.03) improve the ones with FedAvg. On CIFAR-100, the end-to-end+FedProx improves (53.36) over FedAvg (52.54). On the other hand, our method with FedProx does not improve (53.25 vs. 52.3) while remaining comparable to FedAvg. These results may indicate that our method is compatible with optimizations beyond FedAvg.

---

> > ### Comment · Reviewer_1N8v · 2021-11-25
> > **Several concerns still need  to be solved.**
> >
> > After reading the authors' response,  my concerns still exist.
> >
> > Based on the reported experiments, the hyperparameter "S" highly influences the final performance, which makes the proposed ProgFed impractical.
> >
> > In addition, the main theorem can not reflect the influence of hyperparameter "S".  If we can set an appropriate hyperparameter "S"  according to some theoretical instructions, the proposed algorithm would be more appealing.
> >
> > On the other hand, can the authors give more analysis on why the FedProx+ProgFed is worse than End-to-end + FedProx?  Based on the current experiment, I doubt the effectiveness of applying the progressive training technique for federated learning.  Thus, how about incorporating ProgFed with Scaffold and FedAdam?  From the reader's perspective,  they may want to know whether the proposed progressive training technique is suitable for most of the existing federated optimizers.
> >
> > Based on the current response, I would like to keep my initial score.

---

> > > ### Author Response · Authors · 2021-11-29
> > > **More experiments on ablation study and FedAdam**
> > >
> > > We thank Reviewer 1N8v again for the thoughtful comments, which indeed improves the paper. We add more experiments and note that many concerns could be mitigated by hyper-parameter search, as done in prior work [1]. We hope our reply can address the reviewer's concerns.
> > >
> > > > The hyperparameter "S" highly influences the final performance, which makes the proposed ProgFed impractical.
> > >
> > > We find that the performance on CIFAR-100 significantly improves if we slightly increase the total number of epochs from 3000 to 4000: 53.80 (S=3), 54.25 (S=5), 54.46 (S=8). Although the total cost increases as the number of total epochs increases, we make the following observations. (1) Adding too many parameters at a time (e.g., S=2) might hurt the performance (49% even with 4000 epochs). (2, Table 1) The final performance is much higher than the end-to-end training and ProgFed with S=4. (3, Table 2) To fairly compare the settings, we report the communication costs when achieving the final performance of the baseline (52.54). It is observed that ProgFed achieves the same performance while consuming much fewer costs. (4, Table3)  We show that ProgFed can achieve higher performance given the same amount of costs as the number of stages increases.
> > >
> > > Lastly, we note that although we usually adopt the same setting (e.g., number of epochs) from end-to-end training to ProgFed, we believe it is reasonable and practical to optimize hyper-parameters for ProgFed. On the other hand, it also indicates that our method can work well by inheriting the parameters from the end-to-end training in most cases, thus reducing efforts for tuning.
> > >
> > >
> > > Table 1. Ablation study on the number of stages using FedAvg
> > >
> > > |               | Accuracy (%) | Epochs | Cost (GB) |
> > > |---------------|:------------:|:------:|:---------:|
> > > | End-to-End    |     52.54    |  3000  |    5368   |
> > > | ProgFed (S=3) |     53.80    |  4000  |    5695   |
> > > | ProgFed (S=4) |     53.23    |  3000  |    4179   |
> > > | ProgFed (S=5) |     54.25    |  4000  |    5479   |
> > > | ProgFed (S=8) |     54.46    |  4000  |    5166   |
> > >
> > > Table 2. Communication costs to achieve the baseline performance (52.54) using FedAvg.
> > >
> > > |               | Cost |
> > > |---------------|:----:|
> > > | End-to-End    | 5368 |
> > > | ProgFed (S=3) | 4365 |
> > > | ProgFed (S=4) | 3804 |
> > > | ProgFed (S=5) | 3781 |
> > > | ProgFed (S=8) | 3421 |
> > >
> > > Table 3. Accuracy when consuming the same cost (4179 GB) as S=4
> > >
> > > |               | Accuracy (%) |
> > > |---------------|:------------:|
> > > | End-to-end    |     51.19    |
> > > | ProgFed (S=3) |     52.33    |
> > > | ProgFed (S=4) |     53.23    |
> > > | ProgFed (S=5) |     53.56    |
> > > | ProgFed (S=8) |     53.50    |
> > >
> > > > Experiments on ProgFed + FedAdam
> > >
> > > We extend ProgFed to FedAdam, where we disable momentums on clients, set the server learning rate as 0.01, and apply beta1=0.9, beta2=0.99 for Adam. We observe that both the baseline (56.21) and our method (60.55) significantly outperform the ones with FedAvg (52.54 and 53.23, respectively). To further verify the stability, we run the experiment over three random seeds. The result is shown in the following table.
> > >
> > > Table 4. The results on our method and the baseline with FedAdam over three seeds.
> > >
> > > | FedAdam       |       |       |       |       |      |
> > > |---------------|-------|-------|-------|-------|------|
> > > |               | Seed1 | Seed2 | Seed3 |  Mean |  Std |
> > > | End-to-End    | 56.21 | 55.68 | 56.08 | 55.99 | 0.28 |
> > > | FedProg (S=4) |   61  | 60.01 | 60.64 | 60.55 | 0.50 |
> > >
> > > Along with the experiments on FedProx, we have shown that our method generalizes beyond FedAvg, including the server-side variant (FedAdam) and client-side variant (FedProx).
> > >
> > > > Can the authors give more analysis on why the FedProx+ProgFed is worse than End-to-end + FedProx?
> > >
> > > It might be due to hyper-parameter searching. As discussed in Figure 6 in [1], federated optimization methods are sensitive to hyperparameters and often rely on grid search. Due to the time limit, we only conducted a brief experiment on FedAdam. We find that the server learning rate severely affects performance, as shown in the following table. Note that similar to the FedAvg experiments in the main paper, we adopt cosine learning decay.
> > >
> > >
> > > Table 5. Results on FedAdam with different server learning rates
> > >
> > > |               |             lr=1 |           lr=0.1 |  lr=0.01 |
> > > |-----------|--------------:|--------------:|------:|
> > > | Baseline  |      1.00     |      6.00     | 56.21 |
> > > | Ours      | not converged | not converged | 60.55 |
> > >
> > > However, we also show in our experiments that our method with S=4 can directly inherit the learning rate setting from the end-to-end training for most cases and produce comparable performance while consuming fewer costs.
> > >
> > > [1] Reddi, Sashank J., et al. "Adaptive Federated Optimization”, ICLR. 2021.

---

> > > ### Author Response · Authors · 2021-11-29
> > > **More results on EMNIST**
> > >
> > > We additionally present the results on EMNIST with FedAvg, FedProx, and FedAdam. The experiment further supports the applicability of our method. We hope the result can help mitigate the reviewer's concern about practicability.
> > >
> > > Table 6 shows that advanced optimizations indeed improve both the baseline and our method. Despite the comparable performance, our method additionally saves around 29.49% communication costs, as reported in Table 3 (in the main paper).
> > >
> > > Table 6. EMNIST results using FedAvg, FedProx, and FedAdam.
> > >
> > > |               | FedAvg | FedProx | FedAdam |
> > > |:-------------:|:------:|:-------:|:-------:|
> > > | End-to-end    |  85.75 |  86.36  |  86.53  |
> > > | FedProg (S=4) |  85.67 |  86.08  |  86.13  |

---

### Official Review · Reviewer_3Huv · 2021-11-01

**Correctness:** 3
**Technical Novelty And Significance:** 3
**Empirical Novelty And Significance:** 3
**Recommendation:** 6
**Confidence:** 3

**Main Review:**

# Novelty & Significance:

- The idea of progressive training is not new, and the authors cite in particular Karras et al 2018. Given the importance of the topic in the paper, I would have appreciated a paragraph devoted to it in the related work section.
- Theoretical results follow a previous proof structure but seem to bring some novelty. Their significance is not that big insofar as the paper is concerned with deep learning, for which convergence rates do not play a huge role (in particular, the learning rate schedule is chosen independently of the rate derived). Further, the final result is that the convergence is at worst twice slower than the baseline, which is a simple consequence of the choice of spending at least half of the training budget on the full network.
- The experimental results show that the idea can bring some computational and communication savings, in a variety of contexts (cross-device/silo), and is compatible with other approaches.
    - Re computational gains, the benefits seem highly dependent on the architecture (cf fig 3), which reduces its significance.
    - The proposed approach seems only to bring very modest gains with respect to a « random » approach where one would train random sub-networks (cf Table 5). This reduces the significance of the results.

# Writing & Clarity:
- The paper is well written and the main idea is way conveyed.
- However, notations could be simplified (e.g. page 4 the notation $x_{t, E_s^c}$ is introduced but never used) to facilitate understanding
- I would suggest using a log range for the compression factor axis on figs 6a and 6b.

Here are a few minor typos:
- Page 4, paragraph training of progressive model, there is a typo for $T_S$, I think the correct value should be $T (S+1) / 2S$
- Circular notations for the theoretical analysis (end of page 4) for $s = \min(S, t/T_s)$
- End of sec 4.5, missing « is » easy to implement
- Proof of lemma 2, page 13, the first update equation is probably missing a $\gamma_t$ before $g_t^s$

# Quality & Correctness:

## Theoretical results:

- I did not check the details of the proof of Theorem 1.
- Regarding the results, the choice of learning step $\alpha_t$ seems to be hardly usable in practice, as it relies on the knowledge of both the full gradient and the gradient of the sub-network.

## Empirical results

Computation reduction;
- I am not sure to completely understand the metric used in the x-axis of Fig 2, in terms of unit used. Is it the total number of FLOP (i.e. the number of floating point operations over time integrated along time)?
- Why isn’t BraTS reported as well for the computation cost reduction benchmark?

Communication reduction:

- Which architecture is used for the results reported in Fig 4 and 5 for the classification tasks? Additional results are reported in Appendix for varying architectures, but it would be nice to precise it in the main text
- For multiple experiments, the authors average results over 3 seeds, which is a good effort towards reproducibility. Would it be possible to perform 5 runs in order to get stds ? Otherwise, to have access to the individual values?
- There are some instabilities at each step transition (cf Fig 5), and I suspect that warm-up seems to play some role here. The paper is silent about this aspect there were no ablation studies or investigation on this particular ingredient. I would have appreciated a focus on this part rather than e.g.

**Summary Of The Paper:**

This paper introduces the idea of progressive training of deep networks for federated learning, where only some layer blocks are trained initially and the remaining blocks are progressively added, in order to reduce both local computation cost and communication cost in the early stages (as there are fewer weights). The authors provide a convergence proof of this approach, showing the same asymptotic convergence rate as plain training.
Numerical experiments are conducted in EMNIST, CIFAR-10/100 and BraTS, split either in an IID or non-IID fashion and representing either a cross-device or cross-silo setting. Numerical experiments show a highly architecture-dependent reduction in the computation cost in the centralised setting. In the federated setting, communication gains significant communication gains are also achieved. Finally, the authors show that progressive training is compatible with other compression methods, increasing the gain in communication cost.

**Summary Of The Review:**

This paper introduces an existing scheme into the federated setting, and provides an experimental proof of the gains it brings in terms of local computations and communication. Despite some minor issues which can be fixed, I think this paper could be accepted.

---

> ### Public Comment · ~Guoliang_Cheng1 · 2021-11-10
> **Issue on Figure 2**
>
> This is a very meaningful work, but I am also confuse about the metric used in the x-axis of Fig 2. The metric used in the x-axis of Fig 2 look like total number of FLOPs on inference (not training), so it does not represent the computational cost in training process.
>
> Take the first subplot of Fig 2 as example, the computational cost of Resnet18 on a single CIFAR100 sample is 0.08 GFLOPs. To complete a single epoch in centralized setting( Fig 2 setting), the inference cost is  4000 GFLOPs (50000*0.08) approximately. To my understanding, the metric used by the author is inference cost rather than the training cost, it look like run 15 epoch (60000 GFLOPs) in the first subplot of Fig 2. To verify this, I set the x-axis to represent inference cost, and similar curve can be obtained within 15 epoch. Due to the image preprocessing (data augmentation) and higher memory access costs in the training phase, the training costs and inference costs are not linear for different model architectures. I recommend using metrics such as power to measure computational cost.

---

> > ### Author Response · Authors · 2021-11-20
> > **Clarifying the measurement of computation costs**
> >
> > Dear Guoliang,
> >
> > Thanks for the comment.
> >
> > The x-axis of Figure 2 is the total number of FLOPs accumulated over time. Since we train the model with batch size 128, we measure FLOPs in the same manner. For instance, ResNet18 takes 0.34, 0.56, 0.73, and 0.89 GFLOPs on our A100 machines to process one batch with respect to s = 1, 2, 3, 4 when S=4. We do not consider data augmentation and data access since they are independent of architectures.

---

> ### Author Response · Authors · 2021-11-20
> **Reply to Reviewer 3Huv: improving representations and clarifying experiment settings**
>
> We appreciate Reviewer 3Huv's constructive feedback and address the specific comments below.
>
> > A paragraph devoted to progressive learning in the related work section.
>
> We have included a new paragraph in Section D for progressive learning.
>
> > The proposed method brings limited gains over the random strategy.
>
> We re-run the experiment in Table 5 over three random seeds to get the standard deviation. The numbers over 3 seeds are: 73.78, 73.86, 75.5. Besides, the random strategy often requires additional sampling and indexing, causing unnecessary overhead to the system. As shown in Figure 5, the model performance might be affected in a few epochs right after switching stages. Our empirical results suggest that momentums from different stages in federated learning on CIFAR-100 may interfere with the training (53.25% to 51.43%). The clients might have to keep momentums for training different stages, while we can discard those terms after extending models. The random strategy also introduces unnecessary warm-up costs. Thus, ProgFed is favorable over the random strategy.
>
> > Regarding the metric used in the x-axis of Fig 2, is it the total number of FLOP?
>
> Yes. Similar to Figure 5, we report the total number of FLOPs accumulated over time.
>
> > Why isn’t BraTS reported as well for the computation cost reduction benchmark?
>
> We have added the result in Figure 10 and discuss it in Section C.2. Interestingly, we find that despite the fewer parameters required by the symmetric strategy, it takes more computation costs than the asymmetric strategy. This is probably because the former retains the full resolution of feature maps.
>
> > Which architecture is used for the results reported in Fig 4 and 5 for the classification tasks?
>
> For federated experiments, we adopt a ConvNet for EMNIST (Reddi et al., 2021), ResNet-18 for CIFAR-10 (McMahan et al., 2017), and CIFAR-100 (Reddi et al., 2021), and 3D-Unet for BraTS (Sheller et al, 2020). We described more implementation details in Section. 4.1 and Section B.
>
> > Would it be possible to perform 5 runs in order to get stds? Otherwise, to have access to the individual values?
>
> We have added the standard deviation over 3 seeds in the main paper and additionally include the raw numbers in Table 7, 8, and 9.
>
> > There are some instabilities at each step transition (cf Fig 5), and I suspect that warm-up seems to play some role here. A focus on this part would be appreciated.
>
> Yes. Warm-up plays an important role in ProgFed as we discussed in the practical considerations in Sec 3.1. The newly-added layers are randomly initialized and take time to recover the performance, especially in federated learning. For instance, Federated ResNet-18 on CIFAR-100 w/ warm-up achieves 53.23 while w/o warm-up only achieves 51.09.
>
> > Editorial comments and typos
>
> We have corrected $T_S$ and fixed the typos, but $s=min(s, \lceil \frac{t}{T_s} \rceil)$ seems correct.

---

### Public Comment · ~Chaoyang_He1 · 2021-11-09
**An important related works in ICML 2021: PipeTransformer: distributed training with progressive freeze training**

Dear Authors,

In this year's ICML, we've published a progressive training idea in distributed training setting. We believe the idea is highly similar. As for the application of our idea to FL, we already highlight it in our future work.

[PyTorch Blog]
https://pytorch.org/blog/pipetransformer-automated-elastic-pipelining/
(progressive training is introduced)

[ICML proceeding]
http://proceedings.mlr.press/v139/he21a/he21a.pdf

It's highly appreciated if you could discuss our paper as related works. Thanks.

---

> ### Author Response · Authors · 2021-11-20
> **Related work**
>
> Dear Chaoyang,
>
> We thank you for the comment and discussion.
>
> We have checked the ICML paper. Both PipeTransformer and ProgFed share the same idea of partial training. However, we note that our work differentiates from PipeTransformer by the following points.
> 1. Our method starts from the lower-level layers and gradually extends the sub-network, which better aligns with the notion of "progressive learning" as investigated in PGGAN. On the other hand, PipeTransformer trains the model from the entire network and gradually freezes the lower-level layers.
> 2. Although PipeTransformer investigates employing partial training to reduce costs of training transformer-based models, how to apply progressive learning to general federated learning problems remains open.
>
> We appreciate the input and have included the discussion in Section D.

---

### Author Response · Authors · 2021-11-20
**General Reply**

We thank reviewers and commenters for all of their constructive comments. We are glad that reviewers found our paper interesting (R1N8v), compatible with traditional compression (R3Huv, R5a4S, R3zVc), easy but practical (R5a4S), validated with extensive experiments (R3Huv, R5a4S), and well-written (R3Huv, R5a4S).

We have revised the paper in the following points.
- We add standard deviation in Table 2, 3, 5 and leave the standard deviation of Table 4 in Table 10 in the appendix due to the space limit.
- We include Table 7, 8, 9 in the appendix to show the raw numbers of Table 2, 3.
- We add Figure 10 to show the computation costs vs. accuracy on the BraTS dataset and discuss the result in Section C.2.
- We re-run the experiment in Table 5 to compute the standard deviation over three random seeds.
- We include a new paragraph in Section D for related work on progressive learning and mention it at the beginning of Section 2.
- We fix the typos mentioned by the reviewers and add a few sentences about the dataset splits in Section 4.1.

---

> ### Author Response · Authors · 2021-11-22
> **Uploading a new version of the revision**
>
> Dear Reviewers,
>
> We have uploaded a new version of the revision. In particular,
>
> 1. We made all revised parts in blue for your convenience.
> 2. We added descriptions and new results on w/ and w/o warm-up in Section 3.1 so that readers can better understand the role of warm-up in federated settings.
> 3. We included a new paragraph in Section C.6, showing that ProgFed can be extended to ProgProx.

---

### Decision · Program_Chairs · 2022-01-20

**Decision:**

Reject

**Comment:**

As pointed out by some reviewers, the proposed method basically puts progressive training in the federated context. The theoretical analysis only concerns the centralized or non-federated setting and thus give no insight or guidelines for progressive training in federated learning. The main advantage of saving communication mainly comes from the simple observation that less parameters are computed and communicated during each round before the full end-to-end stage. However, this may cause extra overhead in hyper-parameter tuning including number of stage, learning rate schedules and stage-wise warmup. Despite its potential effectiveness in practice, the current version of the paper falls short of the acceptance bar due to the weakness in novelty and relevant theory for federated learning.